# MEAN FLOW POLICY WITH INSTANTANEOUS VELOCITY CONSTRAINT FOR ONE-STEP ACTION GENERATION

**Guojian Zhan**[1,2]*, **Letian Tao**[1]* , **Pengcheng Wang**[2], **Yixiao Wang**[2], **Yiheng Li**[2], **Yuxin Chen**[2],
**Hongyang Li**[3], **Masayoshi Tomizuka**[2], **Shengbo Eben Li**[1]†

[1] School of Vehicle and Mobility & College of AI, Tsinghua University
[2] Berkeley AI Research (BAIR), UC Berkeley
[3] The University of Hong Kong

## ABSTRACT

Learning expressive and efficient policy functions is a promising direction in reinforcement learning (RL). While flow-based policies have recently proven effective in modeling complex action distributions with a fast deterministic sampling process, they still face a trade-off between expressiveness and computational burden, which is typically controlled by the number of flow steps. In this work, we propose mean velocity policy (MVP), a new generative policy function that models the mean velocity field to achieve the fastest one-step action generation. To ensure its high expressiveness, an instantaneous velocity constraint (IVC) is introduced on the mean velocity field during training. We theoretically prove that this design explicitly serves as a crucial boundary condition, thereby improving learning accuracy and enhancing policy expressiveness. Empirically, our MVP achieves state-of-the-art success rates across several challenging robotic manipulation tasks from Robomimic and OGBench. It also delivers substantial improvements in training and inference speed over existing flow-based policy baselines.

## 1 INTRODUCTION

A promising topic in reinforcement learning (RL) community is to develop expressive and efficient policies, particularly in complex control environments where action distributions can be multi-modal (Zhu et al., 2023; Wang et al., 2023). Generative policies, such as diffusion model and flow matching, have recently emerged as a powerful alternative to Gaussian or mixture policies by transforming simple base distributions into flexible action distributions via learnable transformations (Song et al., 2020; Chi et al., 2023). However, a key limitation of existing generative policies is their dependence on iterative multi-step refinement from noise to actions (Wang et al., 2024a; 2025; Ding et al., 2024). This computational dependency imposes a significant overhead that hinders training speed, particularly for online RL where action sampling is a per-step requirement (Li, 2023; Yang et al., 2023). Moreover, this overhead translates to considerable inference latency, which is a major impediment to achieving high closed-loop performance in real-time control systems (Zhan et al., 2024; 2025; Jiang et al., 2024).

A question naturally arises: *Can we unify the expressiveness of generative policies with the efficiency of one-step action generation for online RL?*

In this paper, we propose the mean velocity policy (MVP) as an affirmative answer. While existing flow policies learn instantaneous velocities and require multistep iterative sampling (Lipman et al., 2023; Park et al., 2025; Bharadhwaj et al., 2024), MVP instead learns the mean velocity field (Geng et al., 2025a). This design

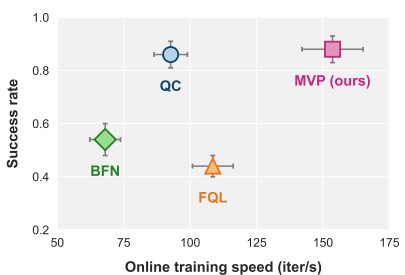

Figure 1: **Performance-efficiency comparison on 9 robotic manipulation tasks.**

---

*Equal contribution.
†Corresponding author. lishbo@tsinghua.edu.cn.

enables a direct, single-step mapping from a base Gaussian noise to a multi-modal action distribution, thereby preserving the expressive power of flow-based models while drastically improving training and inference efficiency (Kornilov et al., 2024).

Although the time-efficiency gains of MVP are very promising, its learning difficulty is higher than that of a standard flow policy. One reason is that our MVP requires modeling the mean velocity for any time interval specified by two time points (Geng et al., 2025a). A more significant reason is that its learning process is governed by a first-order ordinary differential equation (ODE) derived from the definition of mean velocity. However, this ODE theoretically suffers from the problem of multiple solutions due to a lack of explicit boundary conditions, that is, the value at any boundary point is not enforced. This poses a non-trivial challenge to learning accuracy and consequently affects policy expressiveness (Birkhoff & Langer, 1923).

To address this, we introduce an instantaneous velocity constraint (IVC) to compensate for the lack of boundary conditions. Intuitively, IVC pairs the average velocity loss for each interval with an instantaneous velocity loss at the interval's start point. In practice, IVC is implemented as a auxiliary policy loss, adding negligible computational overhead while materially improving accuracy. We evaluate our MVP on Robomimic (Mandlekar et al., 2021) and OGBench (Park et al., 2024), two demanding robot manipulation benchmarks. As shown in Figure 1 and Table 3, MVP achieves state-of-the-art success rates while delivering substantial speed-ups in training and per-step inference over strong flow-policy baselines, on average across both suites.

Our contributions are summarized threefold:

- We propose a new flow-based policy, namely mean velocity policy (MVP), that enables fastest one-step action generation. By modeling the mean velocity field, MVP retains the expressiveness of generative policies while eliminating multi-step sampling overhead.

- We design a training enhancement technique, namely instantaneous velocity constraint (IVC), to improve the learning accuracy of mean velocity field. This technique explicitly serves as a boundary condition, thereby stabilizing learning and enhancing policy expressiveness.

- We empirically achieve state-of-the-art success rates on two challenging robotic manipulation benchmarks: Robomimic and OGBench. Moreover, our approach provides a substantial speedup in both training and inference over existing flow-policy baselines, highlighting its practicality for real-time application.

## 2 PRELIMINARIES

**Reinforcement Learning.** We consider an agent interacting with an environment modeled as a Markov Decision Process (MDP), defined by a tuple $\mathcal{M} = \langle \mathcal{S}, \mathcal{A}, \mathcal{P}, r, \gamma \rangle$ (Li, 2023). The components are the state space $\mathcal{S} \subseteq \mathbb{R}^n$, the action space $\mathcal{A} \subseteq \mathbb{R}^m$, the state transition function $\mathcal{P}(s'|s, a)$, the reward function $r(s, a)$, and the discount factor $\gamma \in [0, 1)$. The primary goal in reinforcement learning (RL) is to learn a policy $\pi(a|s)$ that maximizes the expected cumulative discounted reward, namely return, given by

$$J_\pi = \mathbb{E}_{\pi, \mathcal{P}} \left[ \sum_{k=0}^{\infty} \gamma^k r(s_k, a_k) \right]. \tag{1}$$

Grounded in the off-policy learning paradigm, our approach utilizes an action-value function (Q-function) to guide policy improvement, which denotes the expected cumulative return for taking an action $a$ in a state $s$ and thereafter following the policy $\pi$.

$$Q^\pi(s, a) = \mathbb{E}_{\pi, \mathcal{P}} \left[ \sum_{i=0}^{\infty} \gamma^i r(s_i, a_i) | s_0 = s, a_0 = a \right]. \tag{2}$$

The optimal action-value function, $Q^*(s, a)$, represents the maximum expected return achievable from state $s$ by taking action $a$. The optimal policy $\pi^*$ can then be found by selecting the action that maximizes this function: $\pi^*(s) = \arg\max_{a \in \mathcal{A}} Q^*(s, a)$.

**Flow Matching.** Flow matching is a principled methodology for constructing continuous-time generative models (Lipman et al., 2023). In contrast to diffusion models, which employ stochastic differential equations (SDEs) (Song et al., 2020), flow matching is built upon deterministic dynamics governed by an ordinary differential equation (ODE). By directly learning a continuous-form instantaneous vector field, this approach simplifies the training objective and enables more efficient sampling. Specifically, it trains a neural network $v_\theta : \mathbb{R}^d \times [0, 1] \to \mathbb{R}^d$ to parameterize a velocity field $v(x(t), t)$ that matches a predefined conditional target velocity $v(1)$.

For a source distribution $q(x(0))$ and a target distribution $p(x(1))$, the velocity field is trained by minimizing the flow matching loss (Lipman et al., 2024):

$$\mathcal{L}_{\text{FM}}(\theta) = \mathbb{E}_{\substack{t \sim \mathcal{U}([0,1]) \\ x(1) \sim p, \, x(0) \sim q}} \left\| v_\theta\big(x(t), t\big) - v(x(t), t) \right\|_2^2, \tag{3}$$

where any intermediate point along the generating path $x(t) = tx(1) + (1 - t)x(0)$ is a linear interpolation between source and target points. The velocity for this path is assumed to be a constant vector $v(x(t), t) = x(1) - x(0)$. This formulation defines the target vector field along a straight path between the source and target samples. Once trained, the learned field $v_\theta$ defines a probability flow via the following ODE:

$$\frac{dx(t)}{dt} = v_\theta(x(t), t), \; x(0) \sim q, \tag{4}$$

which allows us to effectively generate samples of the target distribution $p$, starting from samples of the source distribution $q$. Although flow matching is conceptually designed to generate along a straight line, the paths still curve in practice when fitting between two distributions (Song et al., 2023). Therefore, multi-step discretization and numerical methods like the Euler method are often required to solve the ODE in Eq. (4) to obtain high-quality generated results (Park et al., 2025).

## 3 METHOD

First, we introduce the mean velocity policy (MVP), showing how its integration with a "generate-and-select" mechanism enables a direct mapping from noise to optimal actions. We then present the instantaneous velocity constraint (IVC) and theoretically justify its role in improving the learning accuracy. Finally, the complete pseudo-code for our mean flow RL algorithm is provided.

### 3.1 MEAN VELOCITY POLICY

In RL, a policy $\pi(\cdot|s)$ defines a distribution over actions given a state $s$. For standard flow-based policies, this mapping is framed as a generative process: a velocity model, $v(a(t), t, s)$, transforms a standard Gaussian noise (source) into the optimal action (target), with the state serving as a conditioning input. The final output action $a(1)$ is calculated by

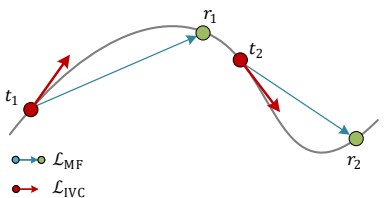

$$a(1) = a(0) + \int_0^1 v(a(\tau), \tau, s)d\tau, \; a(0) \sim \mathcal{N}(0, I). \tag{5}$$

Unlike standard flow policies that learn the instantaneous velocity field $v(a(t), t, s)$, we center on the mean velocity field $u(a(t), t, r, s)$, modeling the mean velocity over any given time interval $[t, r]$ as

Figure 2: Velocity field: blue arrows denote the mean velocity over a time interval, with red arrows representing the instantaneous velocity at a time point.

$$u(a(t), t, r, s) \triangleq \frac{1}{r - t} \int_t^r v(a(\tau), \tau, s)d\tau. \tag{6}$$

The relationship between these two types of velocities is shown in Figure 2. If the mean velocity model $u^*$ is ideally learned, the policy inference is formulated as

$$a(1) = a(0) + u^*(a(0), 0, 1, s), \; a(0) \sim \mathcal{N}(0, I). \tag{7}$$

**(1) How to imitate a given target action.** To train the mean velocity model $u$, we first multiply both sides of Eq. (6) by $(r - t)$ and then differentiate with respect to $t$ ( treating $r$ as independent of $t$), which yields the mean flow identity:

$$-u(a(t), t, r, s) + (r - t)\frac{d}{dt}u(a(t), t, r, s) = -v(a(t), t, s). \tag{8}$$

We assume there is a target optimal action $a^*$, serving as $a(1)$ for imitation. For two randomly sampled time points, $t$ and $r$ (with $t < r$), the intermediate point $a(t)$ is defined by the linear interpolation: $a(t) = t \cdot a(1) + (1 - t) \cdot a(0)$, and $v = a^* - a(0)$. Let $\theta$ denote the learnable parameters, the training objective is to minimize the residual of the mean flow identity in Eq. (8) as

$$\mathcal{L}_{\text{MF}}(\theta) = \mathbb{E}_{t, r < t, a(t)} \left\| u_\theta(a(t), t, r, s) - \text{sg}\left(v - (t - r)\frac{d}{dt}u_\theta(a(t), t, r, s)\right) \right\|_2^2, \tag{9}$$

where "sg" denotes the stop-gradient operator to stabilize training. Calculating this training loss requires computing the total time derivative term, $\frac{d}{dt}u$. Using the chain rule, this term is expanded as shown in Eq. (10), and its computation can be performed efficiently using the Jacobian-vector product (JVP) in modern automatic differentiation libraries.

$$\frac{d}{dt}u_\theta(a(t), t, r, s) = v(a(t), t, s) \cdot \frac{\partial}{\partial a}u_\theta(a(t), r, t, s) + \frac{\partial}{\partial t}u_\theta(a(t), r, t, s), \tag{10}$$

By minimizing Eq. (9), the mean velocity model $u$ can effectively transform a standard Gaussian noise into the desired target action distribution. In RL, however, there is no ground-truth dataset of optimal actions to imitate. Next, we will detail how to gradually find better actions and eventually approach the optimal target action.

**(2) How to find the optimal target action to imitate.** Finding the optimal action $a^*$ directly is not realistic. However, we can use the Q-function to progressively find better actions as imitation targets, and eventually find the optimal action $a^*$ in a bootstrap manner. This mechanism is called "generate-and-select" or "best-of-$N$".

In practice, at any given state $s$, the agent first generate $N$ diverse candidate actions as

$$a^i = a_k^i(1) = \epsilon_i + u_\theta(\epsilon_i, 0, 1, s), \ \ a^i(0) = \epsilon^i \sim \mathcal{N}(0, I), \ i = 1, \cdots, N. \tag{11}$$

Then the critic function $Q_\phi$ parameterized by $\phi$ is employed to evaluate all candidates, and the action yielding the highest Q-value is identified as the final output action for that state:

$$a^\star = \arg \max_{a^i} Q_\phi(s, a^i). \tag{12}$$

We treat the combined mean flow generation process in Eq. (11) and the best-of-$N$ mechanism in Eq. (12) as a unified policy $\pi_\theta^{\text{unified}}$, or simply denoted as $\pi_\theta$. The resulting action, $a^\star$, then serves three purposes: (1) interacting with the environment, (2) acting as the target action for policy training, and (3) calculating the target value for value training.

Although this bootstrap mechanism is intuitive, next we will formally prove that such a imitation-based update guarantees policy improvement by the following theorem.

**Theorem 1** (mean velocity policy Improvement). *Let the new policy $\pi_{new}$ be derived from an old policy $\pi_{old}$ via the $N$-candidate generative update process described above. Under assumptions of a bounded Q-function error ($\epsilon_Q$), $L_Q$-Lipschitz continuity of the Q-function, and a bounded mean flow matching error ($\epsilon_A$), the performance difference between these two policies is lower-bounded by*

$$V^{\pi_{new}}(s) - V^{\pi_{old}}(s) \geq \underbrace{\mathbb{E}_{\tau \sim \pi_{new}}\left[\sum_{t=0}^{\infty} \gamma^t \Delta_N^{\pi_{old}}(s_t)\right]}_{\text{BFN improvement term } \Delta_1} - \underbrace{\frac{2\epsilon_Q + L_Q\epsilon_A}{1 - \gamma}}_{\text{fitting error term } \Delta_2}, \tag{13}$$

*where $V(s)$ is the state-value function, and $\Delta_N^{\pi_{old}}(s)$ is the best-of-$N$ advantage gain, satisfying*

$$\Delta_N^{\pi_{old}}(s) := \mathbb{E}_{a_1, \ldots, a_N \sim \pi_{old}(\cdot|s)}\left[\max_{i=1, \ldots, N} Q^{\pi_{old}}(s, a_i)\right] - V^{\pi_{old}}(s) \geq 0. \tag{14}$$

*Proof.* See Appendix A. □

This theorem decomposes the performance difference into two distinct components: a *BFN improvement term* $\Delta_1$ and a *fitting error term* $\Delta_2$. As we prove in Appendix A.4, the best-of-$N$ advantage gain ($\Delta_N^{\pi_{\text{old}}}$) in $\Delta_1$ is strictly non-negative, implying that the policy improvement can be driven by the benefit of sampling $N$ diverse action candidates. It is also important to note that this improvement is partially counteracted by the fitting error term $\Delta_2$, which stems from the critic's inaccuracy ($\epsilon_Q$) and the mean flow matching error ($\epsilon_A$). This highlights the importance of reducing $\epsilon_A$ to further enhance policy performance. In the next subsection, we will introduce the instantaneous velocity constraint (IVC) to help reduce this fitting error.

## 3.2 THE INSTANTANEOUS VELOCITY CONSTRAINT AS A BOUNDARY CONDITION

As we mentioned above, the underlying principle for training the mean velocity model is Eq. (8), which is a first-order ordinary differential equation (ODE) with respect to $u$. Mathematically speaking, we need to know both the dynamics and at least a boundary condition to accurately solve the unique solution of a ODE. Back to our practical setting, Eq. (8) only provides the dynamics given $t < r$. Although sampling pairs where $r$ is very close to $t$ could implicitly serve as the boundary condition, such events are too rare to ensure robust training.

Inspired by this, we introduce the instantaneous velocity constraint (IVC), a training objective that explicitly enforces a boundary condition at $t$. The mean velocity from $t$ to $t$ is exactly the known instantaneous velocity $v = a^* - a(0)$, so the IVC objective is expressed as:

$$\mathcal{L}_{\text{IVC}}(\theta) = \mathbb{E}_{t,a(t)} \|u_\theta(a(t), t, t) - v\|_2^2. \tag{15}$$

To understand the effectiveness of this IVC objective, we first derive the following theorem, which demonstrates the multiplicity of solutions for Eq. (8) in the absence of a boundary condition.

**Theorem 2** (Multiplicity of Solutions for the Mean Flow Identity). *Let the **cumulative error** $\Delta_u$ be defined as the difference between the learned and the true mean velocity field, $\Delta_u(a(t), t, r) \triangleq u_\theta(a(t), t, r) - u^*(a(t), t, r)$. Assume $u_\theta$ perfectly satisfies the mean flow identity (Eq. (8)) for all $t < r$. Then, its cumulative error $\Delta_u$ satisfies*

$$\Delta_u(a(t), t, r) = \frac{C(a, r)}{r - t} \tag{16}$$

*where $C(a, r)$ is an integration constant independent of time $t$.*

*Proof.* By assumption, both $u_\theta$ and $u^*$ precisely satisfy the mean flow identity. Subtracting the identity for $u^*$ from that of $u_\theta$ yields a homogeneous linear differential equation with respect to the cumulative error $\Delta_u$ as

$$\Delta_u + (t - r)\frac{d}{dt}\Delta_u = 0. \tag{17}$$

By the product rule for derivatives, this is equivalent to $\frac{d}{dt}[(r - t)\Delta_u] = 0$. Integrating with respect to $t$ yields $(r - t)\Delta_u = C(a, r)$, which completes the proof. □

Theorem 2 reveals that any solution learned by perfectly minimizing the $\mathcal{L}_{\text{MF}}$ loss belongs to a family of functions with a unknown constant $C(a, r)$. Because the $\mathcal{L}_{\text{MF}}$ loss is blind to the boundary, it cannot provide a gradient to force $C(a, r)$ to zero. This allows an arbitrary, persistent bias to exist in the learned $u_\theta$. Next, we derive the following theorem to prove that the explicit introduction of IVC eliminates this degree of freedom and restricts the solution space to the unique correct $u^*$.

**Theorem 3** (Uniqueness via the Instantaneous Velocity Constraint). *Let the **boundary error** $\Delta_v$ be defined as the error at the boundary $t$, $\Delta_v(a(t), t) \triangleq u_\theta(a(t), t, t) - v^*(a(t), t)$. Minimizing the IVC loss, $\mathcal{L}_{IVC} = \mathbb{E}[\|\Delta_v\|^2]$, forces the integration constant $C(a, r)$ in Theorem 2 to zero. This eliminates the boundary error and ensures the cumulative error $\Delta_u$ vanishes for all $t < r$.*

*Proof.* The boundary error $\Delta_v$ is the limit of the cumulative error $\Delta_u$ as $r \to t^+$. Applying Eq. (16):

$$\Delta_v(a(t), t) = \lim_{r \to t^+} \frac{C(a, r)}{r - t}. \tag{18}$$

This limit diverges if $C(a, r) \neq 0$. To keep the IVC loss finite, the optimization must prevent this divergence, which requires $C(a, r) = 0$. With $C(a, r) = 0$, the boundary error $\Delta_v$ is zero. Consequently, by invoking Theorem 2, the cumulative error $\Delta_u$ also becomes zero. This completes the proof. □

In essence, Theorem 3 proves that the IVC provides the necessary boundary condition to make the learning problem well-posed. By forcing the error constant to zero at the boundary, the IVC eliminates the cumulative error inherent to the mean flow objective, thereby retaining the high expressive power of the generative model. Moreover, the resulted smaller mean flow matching error $\epsilon_A$ in Theorem 1, which helps enable a more effective policy improvement with each update.

## 3.3 MEAN FLOW REINFORCEMENT LEARNING

This section systematically presents the complete picture of our mean flow RL. The policy training loss $\mathcal{L}_{\text{policy}}$ combines the mean velocity model loss in Eq. (9) with the IVC loss in Eq. (15):

$$\mathcal{L}_{\text{policy}}(\theta) = \mathcal{L}_{\text{MF}}(\theta) + \lambda \mathcal{L}_{\text{IVC}}(\theta), \tag{19}$$

where the balancing hyperparameter $\lambda > 0$ is called IVC coefficient, and the default value is 1.0.

Concurrently, the critic $Q_\phi$ is trained to minimize a standard TD-error in Eq. (20) on transitions $(s_k, a_k, r_k, s_{k+1})$ from the replay buffer, where $k$ is the step index.

$$\mathcal{L}_Q(\phi) = \mathbb{E}\left[\left(Q_\phi(s_k, a_k) - \left(r_k + \gamma Q_\phi(s_{k+1}, a_{k+1}^\star)\right)\right)^2\right]. \tag{20}$$

Recall that we treat the combined mean flow generation process in Eq. (11) and the best-of-$N$ mechanism in Eq. (12) as a unified policy $\pi_\theta$, so the calculation of $a_{k+1}^\star$ also involves a generative-then-select process for ensuring an unbiased policy evaluation.

The overall training scheme decouples the generative policy's imitative training from the Q-value gradient. Instead, the policy improvement is guaranteed by a best-of-$N$ mechanism under the guidance of Q-value selection. This design allows us to leverage the policy's full expressive power while ensuring stable and effective policy improvement. The complete algorithm is shown in Algorithm 1.

---

**Algorithm 1** Mean Flow RL

---

**Input:** mean velocity policy $\pi_\theta$, where $\theta$ is the parameters of $u_\theta$, Critic $Q_\phi$, offline dataset $\mathcal{D}_{\text{offline}}$
Initialize replay buffer $\mathcal{D}$ with $\mathcal{D}_{\text{offline}}$
  ▷ **Phase 1: Offline Pre-training**
**for** offline training step **do**
    Replay a mini-batch from $\mathcal{D}$
    Update policy $\pi_\theta$ by minimizing $\mathcal{L}_{\text{policy}}(\theta)$ with Eq. (19)
    Update critic $Q_\phi$ by minimizing $\mathcal{L}_Q(\phi)$ with Eq. (20)
**end for**
  ▷ **Phase 2: Online Interaction and Fine-tuning**
**for** online training step $k = 1, 2, \ldots$ **do**
    Observe $s_k$, execute $a_k^* = \pi_\theta(s_k)$, receive $s_{k+1}, r_k$, and store $(s_k, a_k^\star, r_k, s_{k+1})$ into $\mathcal{D}$
    Replay a mini-batch from $\mathcal{D}$
    Update policy $\pi_\theta$ by minimizing $\mathcal{L}_{\text{policy}}(\theta)$ with Eq. (19)
    Update critic $Q_\phi$ by minimizing $\mathcal{L}_Q(\phi)$ with Eq. (20)
**end for**

---

# 4 EXPERIMENTS

## 4.1 MAIN EXPERIMENT

**Benchmark.** We consider a total of 9 sparse-reward robotic manipulation tasks with varying difficulties. This includes 3 tasks from the Robomimic benchmark (Mandlekar et al., 2021), `Lift`, `Can` and `Square`, and 6 tasks from OGBench (Park et al., 2024), `cube-double-task 2/3/4` and `cube-triple-task 2/3/4`. For Robomimic, we use the multi-human datasets. For OGBench, we use the default play-style datasets. See more details of these tasks in Appendix C.

**Baselines and our method.** We compare with three latest strong offline-to-online RL baselines. **(1) FQL (flow Q learning)** (Park et al., 2025) first uses behavioral cloning to train a multi-step flow policy on offline data. It then trains a separate one-step policy that imitates the multi-step policy and maximizes Q-values, enabling efficient and stable learning within the data distribution. **(2) BFN (best-of-$N$)** (Ghasemipour et al., 2021) combines the best-of-$N$ sampling with an expressive multi-step flow policy. Specifically, BFN first generates $N$ candidate actions and picks the action (out of $N$) that maximizes the current $Q$-value. **(3) QC (Q-chunking)** (Li et al., 2025) applies action chunking (Bharadhwaj et al., 2024) on the basis of BFN to improve exploration and sample efficiency. **(4) Ours MVP** also adopts the chunking trick, but leverages a more efficient mean velocity policy to achieve the fastest one-step action generation with maintained or even enhanced expressiveness.

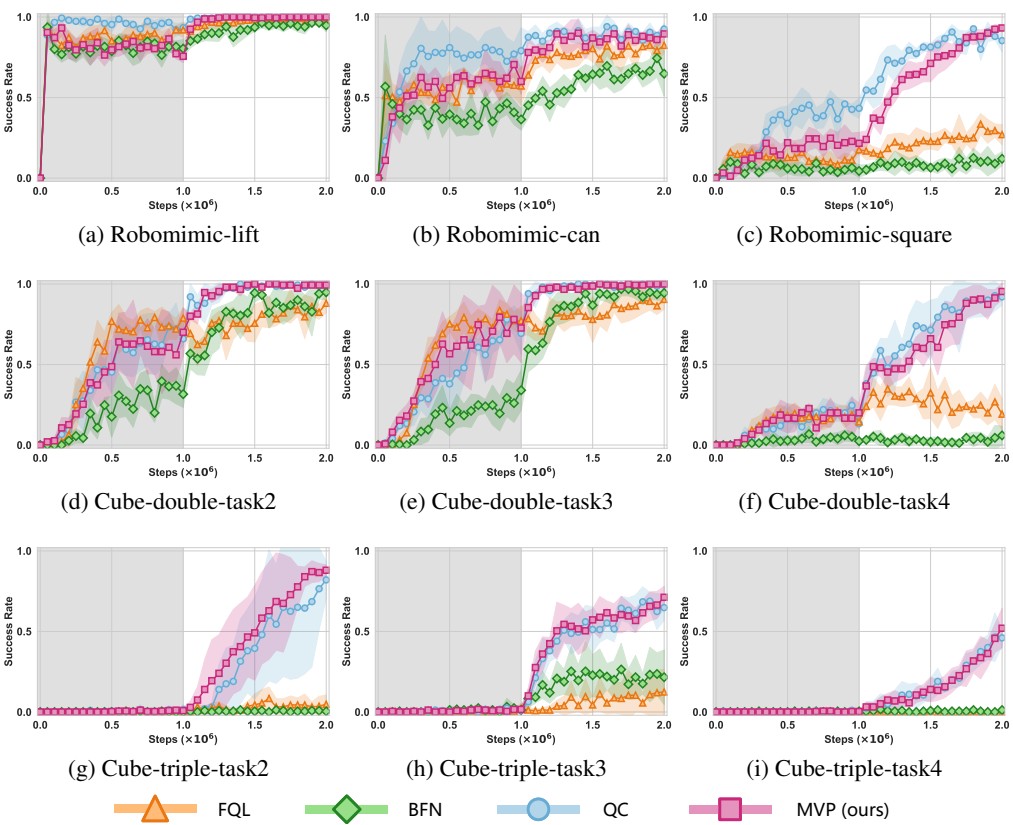

Figure 3: **Training curves on benchmarks.** The solid lines correspond to mean and shaded regions correspond to 95% confidence interval over five runs. The shadow background indicates the offline training phase, while the white background indicates the online training phase.

**Main results.** As shown in Table 1, our MVP matches or exceeds state-of-the-art multi-step flow-matching baselines on eight of nine tasks. On the remaining task, MVP ranks second, with a performance of 0.92, which is just 0.02 points below the top-performing baseline's score of 0.94. While all methods achieve near-perfect performance on simpler tasks like Robomimic-lift, Cube-double-task2, and Cube-double-task3, MVP demonstrates clear superiority on the more challenging tasks. Specifically, MVP consistently outperforms all baselines on Robomimic-square, Cube-double-task4, and all Cube-triple tasks, where it consistently achieves the highest success rates. For instance, on the most difficult task, Cube-triple-task4, MVP achieves a success rate of $0.52 \pm 0.11$, which is significantly higher than the next-best baseline, QC ($0.46 \pm 0.13$), and substantially exceeds both FQL and BFN. Overall, our MVP secures the top position with an average success rate of $0.88 \pm 0.05$. This result highlights its strong capability that is competitive with multi-step flow policies in solving long-horizon, sparse-reward tasks.

Table 1: Success rates. Mean ± Std over 5 seeds. **Bold** = best, underlined = 2nd-best.

| Task | FQL | BFN | QC | MVP (ours) |
|---|---|---|---|---|
| Robomimic-lift | $0.96 \pm 0.03$ | $\mathbf{1.00 \pm 0.01}$ | $\mathbf{1.00 \pm 0.00}$ | $\mathbf{1.00 \pm 0.00}$ |
| Robomimic-can | $0.74 \pm 0.11$ | $0.82 \pm 0.03$ | $\mathbf{0.94 \pm 0.06}$ | $\underline{0.92 \pm 0.07}$ |
| Robomimic-square | $0.12 \pm 0.05$ | $0.34 \pm 0.06$ | $\underline{0.92 \pm 0.01}$ | $\mathbf{0.93 \pm 0.01}$ |
| Cube-double-task2 | $0.95 \pm 0.04$ | $0.88 \pm 0.05$ | $\mathbf{1.00 \pm 0.00}$ | $\mathbf{1.00 \pm 0.00}$ |
| Cube-double-task3 | $0.97 \pm 0.04$ | $0.90 \pm 0.06$ | $\mathbf{1.00 \pm 0.00}$ | $\mathbf{1.00 \pm 0.00}$ |
| Cube-double-task4 | $0.08 \pm 0.04$ | $0.35 \pm 0.09$ | $\underline{0.93 \pm 0.08}$ | $\mathbf{0.95 \pm 0.04}$ |
| Cube-triple-task2 | $0.01 \pm 0.02$ | $0.08 \pm 0.06$ | $\underline{0.82 \pm 0.10}$ | $\mathbf{0.88 \pm 0.03}$ |
| Cube-triple-task3 | $0.12 \pm 0.13$ | $0.26 \pm 0.14$ | $\underline{0.69 \pm 0.05}$ | $\mathbf{0.71 \pm 0.06}$ |
| Cube-triple-task4 | $0.00 \pm 0.00$ | $0.02 \pm 0.02$ | $\underline{0.46 \pm 0.13}$ | $\mathbf{0.52 \pm 0.11}$ |
| Average | $0.44 \pm 0.05$ | $0.52 \pm 0.06$ | $\underline{0.86 \pm 0.05}$ | $\mathbf{0.88 \pm 0.05}$ |

## 4.2 ABLATION STUDY AND SUPPLEMENTARY EXPERIMENTS

**(1) Ablation on the instantaneous velocity constraint (IVC).** We perform an ablation study on the IVC coefficient $\lambda$. Our full version ($\lambda = 1.0$) was compared against variants with a reduced constraint ($\lambda = 0.5$) and without the constraint ($\lambda = 0.0$). The results, as shown in Figure 4, indicate a positive correlation between the IVC weight and performance, while also demonstrating that the method is not overly sensitive to $\lambda$. For example, the success rate on the challenging Cube-triple-task4 significantly increases from $0.30 \pm 0.21$ (with no IVC) to $0.45 \pm 0.15$ (with a partial IVC), and further to $0.52 \pm 0.11$ (with full IVC). Detailed numerical results are listed in Table 4 in Appendix B.1. These findings empirically validate our theoretical claims, confirming the IVC's role as a crucial component for modeling an accurate mean velocity field and consequently achieving significant performance gains.

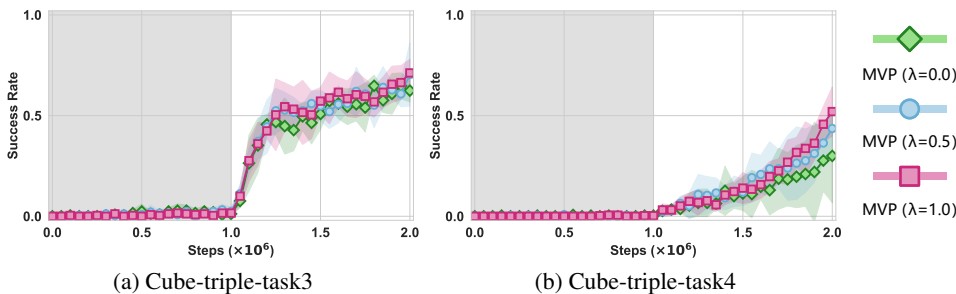

(a) Cube-triple-task3          (b) Cube-triple-task4

Figure 4: **Training curves of ablation on the IVC.**

**(2) Comparison with one-step variants of the aforementioned baselines.** We compared our MVP against one-step variants of the aforementioned baselines: FQL-Onestep, BFN-Onestep, and QC-Onestep. As shown in Figure 5, a naive one-step configuration is insufficient for solving these complex, long-horizon tasks, with baselines achieving success rates near zero on both Cube-triple-task3 and Cube-triple-task4. In stark contrast, our MVP achieves success rates of $0.71 \pm 0.06$ and $0.52 \pm 0.11$ on these tasks, respectively. Detailed numerical results can be seen in Table 5 in Appendix B.1. This supplementary comparison highlights that simply using a one-step standard flow is not enough; the superior expressive capability and stable learning process of our mean velocity policy are critical for tackling these challenging long-horizon manipulation tasks.

**(3) Training and inference time analysis.** Figure 1 presents a comparison of the average success rate (%) versus online training speed (iter/s) across the all 9 tasks. Our MVP achieves highest success rate and fastest training speed. This superior training efficiency stems directly from our one-step action generation, which eliminates the expensive iterative sampling process required by prior multi-step flow policies.

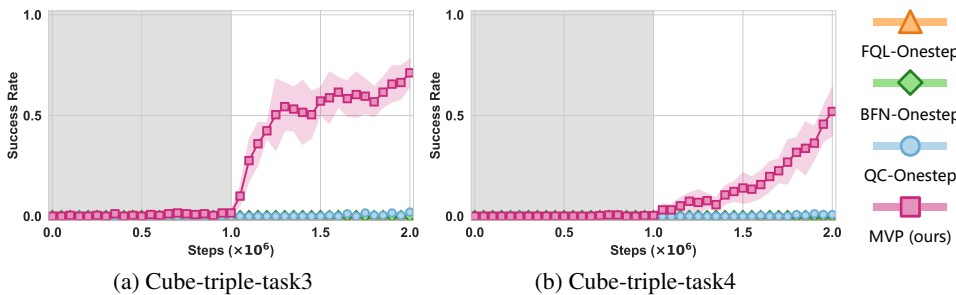

(a) Cube-triple-task3      (b) Cube-triple-task4

Figure 5: **Training curves of comparison with one-step flow.**

Table 2: Comparison of online training speed

| Metric | FQL | BFN | QC | MVP (ours) |
|---|---|---|---|---|
| Average | $108.5 \pm 7.7$ iter/s | $68.0 \pm 5.8$ iter/s | $92.6 \pm 6.3$ iter/s | $153.6 \pm 11.5$ iter/s |

Regarding inference time-efficiency, we conducted an evaluation with a focus on its suitability for real-world robotic deployment. Since robotic platforms often have limited computational resources, our experiments were conducted on a CPU-only environment, AMD Ryzen Threadripper 3960X 24-Core Processor. To simulate a more realistic deployment scenario without hardware acceleration, we disabled JAX's Just-In-Time (JIT) compilation during all evaluations.

The results are listed in Table 3. our MVP and FQL exhibit very similar inference times, with both approaches being significantly faster than BFN and QC.

**Why BFN and QC are slow.**  The poor performance of BFN and QC is primarily because they rely on a 10-step flow policy, which requires iterative computation to transform noise into an action.

**Why FQL is fast.**  FQL simultaneously learns both a 10-step flow policy for high-accuracy imitation and a separate one-step flow policy. The one-step policy is obtained through a distillation process combined with a Q-function maximization loss, which allows FQL to achieve a high inference speed comparable to our MVP.

**Why MVP is better than FQL comprehensively.**  Despite FQL's relatively fast inference, its training process is very slow due to the involvement of multiple policies, including a multi-step flow policy. Furthermore, benchmark results indicate that its success rate is very low, averaging only half of our MVP's. When considering FQL's overall low success rate and slow training speed, our MVP still maintains a significant advantage.

Table 3: Comparison of inference time

| Metric | FQL | BFN | QC | MVP (ours) |
|---|---|---|---|---|
| Average | $10.76 \pm 1.02$ ms | $117.3 \pm 13.23$ ms | $113.22 \pm 11.92$ ms | $10.93 \pm 0.95$ ms |

## 5 RELATED WORK

**Offline-to-online RL.** The offline-to-online RL first uses a static, pre-collected dataset to offline train initial policy and Q-value functions (Levine et al., 2020; Kumar et al., 2020). This process provides a warm start, giving the agent a foundational understanding of the environment that significantly accelerates and improves the efficiency of subsequent online interactive fine-tuning (Lee et al., 2022). Numerous algorithmic designs have been proposed to improve offline-to-online RL, including behavioral regularization (Ashvin et al., 2020; Tarasov et al., 2023), conservatism (Kumar et al., 2020),

in-sample maximization (Kostrikov et al., 2022; Garg et al., 2023), out-of-distribution detection (Yu et al., 2020; Kidambi et al., 2020; Nikulin et al., 2023) and dual RL (Lee et al., 2021; Sikchi et al., 2024). Beyond these algorithmic solutions, the choice of policy function also plays an important role (Wang et al., 2023). A policy network with high expressiveness can better capture the intricate distribution of the behavioral policy during offline training stage. This capability is also crucial for adapting to target environments during online fine-tuning, as optimal actions in these settings often possess a naturally multi-modal nature (Park et al., 2025; Li et al., 2025). Since the policy must infer actions at every step during both the online fine-tuning and real-time deployment stages, the inference time efficiency of the policy function is also critical (Li, 2023; Park et al., 2025). Our work contributes a new policy function, MVP, which achieves the fastest single-step action generation and maintains a high expressive capacity to achieve high performance.

**Generative models as RL policies.** The expressive power of generative models, such as denoising diffusion models (Ho et al., 2020) and flow matching (Lipman et al., 2022), makes them promising for representing complex, multi-modal policies in both offline (Janner et al., 2022; Chi et al., 2023) and online (Yang et al., 2023; Wang et al., 2024a; Ding et al., 2024) RL. However, their iterative sampling process requires a high number of function evaluations (NFE), creating a prohibitive latency for the high-throughput nature of online RL. A common approach in RL to improving time-efficiency of generative policies is distillation, which compresses a trained iterative model into a one-step policy (Wang et al., 2024b; Park et al., 2025). Beyond the field of RL, recent studies have begun to explore training a single-step generative flow directly for tasks like image generation (Lu & Song, 2025). These methods typically operate either by enforcing consistency constraints on the model's outputs at different time steps (Song et al., 2023; Song & Dhariwal, 2024; Geng et al., 2025b) or by explicitly modeling the flow velocity over a specific time interval (Boffi et al., 2024; Frans et al., 2024). The latter class of methods generally requires more iterations to train, but ultimately performs better (Frans et al., 2024). A typical representative, mean flow, has achieved the best one-step fitting performance on Imagenet (Geng et al., 2025a). We propose a new policy function, MVP, which combines mean flow with RL to enable the fastest single-step action generation. In the context of RL, the dynamic shifts in data distribution during sampling place higher demands on imitation-based flow matching training. To address this, our proposed IVC serves as an explicit boundary condition during MVP training, which reduces fitting error and consequently ensures strong policy expressiveness.

## 6 CONCLUSION

We propose the mean velocity policy (MVP), a new generative RL policy that enjoys both high time-efficiency and expressiveness. The former stems from the fastest one-step action generation, and the latter contributes to our designed instantaneous velocity constraint (IVC), which explicitly serves as a necessary boundary condition and reliably improves the learning accuracy. Empirical results on the Robomimic and OGBench benchmarks confirm that our MVP achieves state-of-the-art success rates and offers substantial improvements in training and inference speed. We believe that this work represents a significant step towards developing highly expressive and efficient policy functions for complex robotic control tasks. Regarding limitation, a primary one is the additional GPU memory consumption during training, which stems from the Jacobian-Vector Product (JVP) operation. In future work, we plan to validate our method on more tasks and real robotic platforms.

## 7 ACKNOWLEDGMENT

This study is supported by the Tsinghua University-Toyota Joint Research Center for AI Technology of Automated Vehicle, Beijing Natural Science Foundation (L257002), and SunRisingAI Lab.

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

# A   THEORETICAL ANALYSIS ON THE MEAN VELOCITY POLICY IMPROVEMENT

In this section, we provide the detailed theoretical analysis for the policy improvement guarantee of our training paradigm. It includes the implementation procedures of policy update detailed in A.1, core assumptions and useful lemma in A.2, the formal proof of the mean velocity policy improvement theorem in A.3, and three key properties of the improvement gain in A.4.

## A.1   IMPLEMENTATION PROCEDURES OF POLICY UPDATE

Let $\pi_{\text{old}}$ denote the base mean velocity policy prior to an update. At the same time, we have a learned critic, $Q_\phi(s, a)$, which is an approximation of the true action-value function of the base policy, $Q^{\pi_{\text{old}}}(s, a)$. Let $\pi_{\text{new}}$ denote the updated policy, which is derived from $\pi_{\text{old}}$ through the following three-step generative process for any given state $s$:

1. **Sample**: Generate $N$ candidate actions $\{a_1, \dots, a_N\}$ by sampling from the base policy, i.e., $a_i \sim \pi_{\text{old}}(\cdot|s)$.

2. **Select**: Use the learned critic $Q_\phi$ to select the best action from the candidates, which becomes the target action $a^*(s)$:
$$a^*(s) = \arg\max_{a_i} Q_\phi(s, a_i).$$

3. **Match**: The new policy $\pi_{\text{new}}$ is obtained by training the parameters of $\pi_{\text{old}}$ to match the target action $a^*(s)$. Note that an action $a_{\text{new}} \sim \pi_{\text{new}}$ is not guaranteed to perfectly match $a^*(s)$. We formally bound the expected distance between them in Assumption 3.

For analytical clarity, we introduce a conditional matching distribution, $M(\cdot|a^*)$, to model the outcome of the matching process (Step 3). Specifically, we assume the specific target $a^*$ has been selected in the previous Steps 1 and 2, then generation of the final action $a_{\text{new}}$ is expressed as
$$a_{\text{new}} \sim M(\cdot|a^*).$$
The difference between $M(\cdot|a^*)$ and $\pi_{\text{new}}$ is that $M(\cdot|a^*)$ describes the action distribution conditioned on a specific target action $a^*$, whereas $\pi_{\text{new}}$ represents the marginal distribution of the action, which results from averaging over all possible targets $a^*(s)$ generated in Steps 1 and 2.

## A.2   CORE ASSUMPTIONS AND USEFUL LEMMA

Our analysis relies on the following assumptions and lemma.

**Assumption 1** (Bounded Q-Value Fitting Error). *The error between the learned critic $Q_\phi$ and the true Q-function of the base policy $Q^{\pi_{\text{old}}}$ is uniformly bounded by a constant $\epsilon_Q \geq 0$.*
$$\forall (s, a) \in \mathcal{S} \times \mathcal{A}, \quad |Q_\phi(s, a) - Q^{\pi_{\text{old}}}(s, a)| \leq \epsilon_Q.$$

**Assumption 2** (Q-Value Smoothness). *The true Q-value of the base policy, $Q^{\pi_{\text{old}}}(s, a)$, is $L_Q$-Lipschitz continuous with respect to the action $a$.*
$$\forall s \in \mathcal{S}, \, a_1, a_2 \in \mathcal{A}, \quad |Q^{\pi_{\text{old}}}(s, a_1) - Q^{\pi_{\text{old}}}(s, a_2)| \leq L_Q \cdot d(a_1, a_2).$$

**Assumption 3** (Bounded Mean Flow Matching Error). *The expected distance between the action $a_{\text{new}}$ generated by the new policy and the target action $a^*$ is bounded by a constant $\epsilon_A \geq 0$.*
$$\forall s \in \mathcal{S}, \quad \mathbb{E}_{a_{\text{new}} \sim M(\cdot|a^*)}[d(a_{\text{new}}, a^*(s))] \leq \epsilon_A.$$

**Lemma 1** (Performance Difference Lemma (Kakade & Langford, 2002)). *Let $\mathcal{M} = (\mathcal{S}, \mathcal{A}, P, R, \gamma)$ be a Markov Decision Process, and let $\pi$ and $\pi'$ be two arbitrary policies. The difference in the state-value function at a starting state $s$, or denoted as $s_0$, can be expressed in terms of the advantage function of the new policy $\pi'$ with respect to the old policy $\pi$ as:*
$$V^{\pi'}(s) - V^\pi(s) = \mathbb{E}_{s \sim d^{\pi'}, a \sim \pi'(\cdot|s)} \left[ \sum_{t=0}^\infty \gamma^t A^\pi(s, a) \right].$$

where $d^{\pi'}(s)$ is the discounted state visitation distribution under policy $\pi'$, and $A^\pi(s,a) = Q^\pi(s,a) - V^\pi(s)$ is the advantage function of policy $\pi$.

*Proof.* The proof relies on the key identity $V^\pi(s) = \sum_{a \in \mathcal{A}} \pi(a|s)Q^\pi(s,a)$. We start with the definition of the value difference for a state $s$:

$$V^{\pi'}(s) - V^\pi(s) = V^{\pi'}(s) - \sum_{a \in \mathcal{A}} \pi'(a|s)Q^\pi(s,a) + \sum_{a \in \mathcal{A}} \pi'(a|s)Q^\pi(s,a) - V^\pi(s)$$

$$= V^{\pi'}(s) - \sum_{a \in \mathcal{A}} \pi'(a|s)Q^\pi(s,a) + \sum_{a \in \mathcal{A}} \pi'(a|s)(Q^\pi(s,a) - V^\pi(s))$$

$$= V^{\pi'}(s) - \sum_{a \in \mathcal{A}} \pi'(a|s)Q^\pi(s,a) + \sum_{a \in \mathcal{A}} \pi'(a|s)A^\pi(s,a)$$

Using the Bellman expectation equation for $V^{\pi'}(s)$:

$$V^{\pi'}(s) = \sum_{a' \in \mathcal{A}} \pi'(a'|s) \left( R(s,a') + \gamma \sum_{s' \in \mathcal{S}} P(s'|s,a')V^{\pi'}(s') \right)$$

And the definition of $Q^\pi(s,a)$:

$$Q^\pi(s,a) = R(s,a) + \gamma \sum_{s' \in \mathcal{S}} P(s'|s,a)V^\pi(s')$$

Substitute these into our first equation:

$$V^{\pi'}(s) - V^\pi(s)$$

$$= \sum_{a' \in \mathcal{A}} \pi'(a'|s) \left( R(s,a') + \gamma \sum_{s' \in \mathcal{S}} P(s'|s,a')V^{\pi'}(s') - Q^\pi(s,a') \right) + \sum_{a' \in \mathcal{A}} \pi'(a'|s)A^\pi(s,a')$$

$$= \sum_{a' \in \mathcal{A}} \pi'(a'|s) \left( \gamma \sum_{s' \in \mathcal{S}} P(s'|s,a')(V^{\pi'}(s') - V^\pi(s')) \right) + \mathbb{E}_{a \sim \pi'(\cdot|s)}[A^\pi(s,a)]$$

$$= \gamma \mathbb{E}_{a \sim \pi'(\cdot|s), s' \sim P(\cdot|s,a)}[V^{\pi'}(s') - V^\pi(s')] + \mathbb{E}_{a \sim \pi'(\cdot|s)}[A^\pi(s,a)]$$

This recursive formula shows that the value difference at state $s$ depends on the expected value difference at the next state $s'$. By recursively unrolling this expression over time and averaging over the state visitation distribution $d^{\pi'}(s)$, we arrive at the final result.

$$V^{\pi'}(s) - V^\pi(s) = \mathbb{E}_{s \sim d^{\pi'}, a \sim \pi'(\cdot|s)} \left[ \sum_{t=0}^{\infty} \gamma^t A^\pi(s,a) \right].$$

This completes the proof. □

### A.3 PROOF OF THE MEAN VELOCITY POLICY IMPROVEMENT THEOREM

This subsection provides a formal proof of the mean velocity policy improvement theorem, that is, Theorem 1 in the main text.

*Proof.* Our proof begins by invoking the Performance Difference Lemma, i.e., Lemma 1, which yields

$$V^{\pi_{\text{new}}}(s) - V^{\pi_{\text{old}}}(s) = \mathbb{E}_{\tau \sim \pi_{\text{new}}} \left[ \sum_{t=0}^{\infty} \gamma^t A^{\pi_{\text{old}}}(s_t, a_t) \right]. \tag{21}$$

This equation connects the improvement in value function to the expected advantage of the new policy's actions, measured with respect to the old policy.

The following proof proceeds in two parts: first, we derive the single-step advantage bound $\mathbb{E}_{a_t \sim \pi_{\text{new}}(\cdot|s_t)}[A^{\pi_{\text{old}}}(s_t, a_t)]$, and second, we aggregate them over time to obtain the required bound of value difference, which completes the proof.

**Part 1: Lower bound the single-step expected advantage** We begin by analyzing the Q-value decay due to the imperfect matching for a *given* target action $a^*$. The final action $a_{\text{new}}$ is distributed according to $M(\cdot|a^*)$. We can bound the expected difference as follows:

$$
\begin{aligned}
Q^{\pi_{\text{old}}}(s, a^*) &- \mathbb{E}_{a_{\text{new}} \sim M(\cdot|a^*)}[Q^{\pi_{\text{old}}}(s, a_{\text{new}})] \\
&= \mathbb{E}_{a_{\text{new}}}[Q^{\pi_{\text{old}}}(s, a^*) - Q^{\pi_{\text{old}}}(s, a_{\text{new}})] \\
&\leq \mathbb{E}_{a_{\text{new}}}[|Q^{\pi_{\text{old}}}(s, a^*) - Q^{\pi_{\text{old}}}(s, a_{\text{new}})|] && \text{(by Jensen's inequality)} \\
&\leq \mathbb{E}_{a_{\text{new}}}[L_Q \cdot d(a^*, a_{\text{new}})] && \text{(by Q-Value Smoothness, Assumption. 2)} \\
&\leq L_Q \epsilon_A && \text{(by Bounded Matching Error, Assumption 3)}
\end{aligned}
$$

Rearranging gives us the following bound for a given $a^*$:

$$
\mathbb{E}_{a_{\text{new}} \sim M(\cdot|a^*)}[Q^{\pi_{\text{old}}}(s, a_{\text{new}})] \geq Q^{\pi_{\text{old}}}(s, a^*) - L_Q \epsilon_A.
$$

Now, we take the expectation over the distribution of target actions $a^*(s)$ to get the bound for the marginal policy $\pi_{\text{new}}$:

$$
\begin{aligned}
\mathbb{E}_{a \sim \pi_{\text{new}}}[Q^{\pi_{\text{old}}}(s, a)] &= \mathbb{E}_{\{a_i\} \sim \pi_{\text{old}}}\left[\mathbb{E}_{a_{\text{new}} \sim M(\cdot|a^*(s))}[Q^{\pi_{\text{old}}}(s, a_{\text{new}})]\right] \\
&\geq \mathbb{E}_{\{a_i\} \sim \pi_{\text{old}}}[Q^{\pi_{\text{old}}}(s, a^*)] - L_Q \epsilon_A.
\end{aligned}
$$

Next, we relate the Q-value of the target action, $Q^{\pi_{\text{old}}}(s, a^*)$, back to the Q-values of the original candidates $\{a_i\}$:

$$
\begin{aligned}
Q^{\pi_{\text{old}}}(s, a^*) &\geq Q_\phi(s, a^*) - \epsilon_Q && \text{(by Bounded Q-Value Fitting Error, Assumption 1)} \\
&= \max_{i=1,\ldots,N} Q_\phi(s, a_i) - \epsilon_Q && \text{(by definition of } a^*) \\
&\geq \max_{i=1,\ldots,N} Q^{\pi_{\text{old}}}(s, a_i) - 2\epsilon_Q. && \text{(by Assumption 1 on each candidate } a_i)
\end{aligned}
$$

Substituting this result back into our bound for $\mathbb{E}_{a \sim \pi_{\text{new}}}[Q^{\pi_{\text{old}}}(s, a)]$, we get:

$$
\mathbb{E}_{a \sim \pi_{\text{new}}}[Q^{\pi_{\text{old}}}(s, a)] \geq \mathbb{E}_{\{a_i\} \sim \pi_{\text{old}}}\left[\max_{i=1,\ldots,N} Q^{\pi_{\text{old}}}(s, a_i)\right] - 2\epsilon_Q - L_Q \epsilon_A.
$$

Finally, subtracting $V^{\pi_{\text{old}}}(s) = \mathbb{E}_{a \sim \pi_{\text{old}}}[Q^{\pi_{\text{old}}}(s, a)]$ from both sides gives the desired single-step advantage bound:

$$
\mathbb{E}_{a \sim \pi_{\text{new}}}[A^{\pi_{\text{old}}}(s, a)] \geq \Delta_N^{\pi_{\text{old}}}(s) - 2\epsilon_Q - L_Q \epsilon_A,
$$

where $\Delta_N^{\pi_{\text{old}}}(s)$ is the **best-of-$N$ advantage gain**, defined as

$$
\Delta_N^{\pi_{\text{old}}}(s) := \mathbb{E}_{a_1, \ldots, a_N \sim \pi_{\text{old}}(\cdot|s)}\left[\max_{i=1,\ldots,N} Q^{\pi_{\text{old}}}(s, a_i)\right] - V^{\pi_{\text{old}}}(s).
$$

**Part 2: Extension to the value function difference** With the single-step advantage bound established, we substitute it into Eq. (21) and obtain:

$$
\begin{aligned}
V^{\pi_{\text{new}}}(s) &- V^{\pi_{\text{old}}}(s) \\
&= \mathbb{E}_{\tau \sim \pi_{\text{new}}}\left[\sum_{t=0}^{\infty} \gamma^t A^{\pi_{\text{old}}}(s_t, a_t)\right] \\
&= \sum_{t=0}^{\infty} \gamma^t \mathbb{E}_{s_t \sim d_{s, \pi_{\text{new}}}^t}\left[\mathbb{E}_{a_t \sim \pi_{\text{new}}(\cdot|s_t)}[A^{\pi_{\text{old}}}(s_t, a_t)]\right] && \text{(by law of total expectation)} \\
&\geq \sum_{t=0}^{\infty} \gamma^t \mathbb{E}_{s_t \sim d_{s, \pi_{\text{new}}}^t}[\Delta_N^{\pi_{\text{old}}}(s_t) - 2\epsilon_Q - L_Q \epsilon_A] && \text{(by substituting the bound from Part 1)} \\
&= \mathbb{E}_{\tau \sim \pi_{\text{new}}}\left[\sum_{t=0}^{\infty} \gamma^t \Delta_N^{\pi_{\text{old}}}(s_t)\right] - \frac{2\epsilon_Q + L_Q \epsilon_A}{1 - \gamma}. && \text{(by rearranging the geometric series)}
\end{aligned}
$$

This completes the proof. $\qquad\square$

## A.4 KEY PROPERTIES OF THE BEST-OF-$N$ ADVANTAGE GAIN

The term $\Delta_N^{\pi_{\text{old}}}(s)$ has several important properties:

1. **Non-negativity**: $\Delta_N^{\pi_{\text{old}}}(s) \geq 0$.
2. **Monotonicity with $N$**: $\Delta_{N+1}^{\pi_{\text{old}}}(s) \geq \Delta_N^{\pi_{\text{old}}}(s)$.
3. **Special case** ($N = 1$): $\Delta_1^{\pi_{\text{old}}}(s) = 0$.

*Proof.* Our proof sketch is to first rewrite $\Delta_N^{\pi_{\text{old}}}(s)$ as an integral involving the cumulative distribution function (CDF) of the Q-values, and then demonstrate the claimed properties of the rewritten form.

Let us define a random variable $X$ representing the candidate Q-values. For a given state $s$, the randomness of $X$ is induced by the candidate action $a$ independently sampled from the base policy:

$$X = Q^{\pi_{\text{old}}}(s, a), \quad \text{where } a \sim \pi_{\text{old}}(\cdot|s).$$

Recall that the definition of the **best-of-$N$ advantage gain** is

$$\Delta_N^{\pi_{\text{old}}}(s) := \mathbb{E}_{a_1,\ldots,a_N \sim \pi_{\text{old}}(\cdot|s)} \left[ \max_{i=1,\ldots,N} Q^{\pi_{\text{old}}}(s, a_i) \right] - V^{\pi_{\text{old}}}(s).$$

We can equivalently describe this definition using the CDF notation of the random variable $X$. Let $F_X(x)$ be the CDF of $X$, and $X_1, \ldots, X_N$ be $N$ independent and identically distributed (i.i.d.) samples of $X$, and let $Y_N = \max(X_1, \ldots, X_N)$. The CDF of $Y_N$ is $F_{Y_N}(y) = [F_X(y)]^N$. Therefore, we have $\Delta_N^{\pi_{\text{old}}}(s) = \mathbb{E}[Y_N] - \mathbb{E}[X]$.

Moving forward, since the expectation of a random variable $X$ in terms of its CDF satisfies $\mathbb{E}[X] = \int_0^\infty (1 - F_X(x))dx - \int_{-\infty}^0 F_X(x)dx$, we can further rewrite $\Delta_N^{\pi_{\text{old}}}(s)$ as:

$$\Delta_N^{\pi_{\text{old}}}(s) = \left( \int_0^\infty (1 - F_{Y_N}(x))dx - \int_{-\infty}^0 F_{Y_N}(x)dx \right) - \left( \int_0^\infty (1 - F_X(x))dx - \int_{-\infty}^0 F_X(x)dx \right)$$

$$= \int_{-\infty}^\infty (F_X(x) - F_{Y_N}(x))dx.$$

Substituting $F_{Y_N}(x) = [F_X(x)]^N$, we arrive at the final rewritten form:

$$\Delta_N^{\pi_{\text{old}}}(s) = \int_{-\infty}^\infty (F_X(x) - [F_X(x)]^N)dx.$$

Next we use this form to prove the three key properties of $\Delta_N^{\pi_{\text{old}}}(s)$.

**1. Proof of non-negativity ($\Delta_N^{\pi_{\text{old}}}(s) \geq 0$)**

The CDF $F_X(x)$ takes values in the range $[0, 1]$. For any value $p \in [0, 1]$ and any integer $N \geq 1$, we have $p^N \leq p$. Therefore, the integrand $F_X(x) - [F_X(x)]^N$ is always greater than or equal to zero for all $x$. The integral of a non-negative function is non-negative. Thus, $\Delta_N^{\pi_{\text{old}}}(s) \geq 0$.

**2. Proof of monotonicity with $N$ ($\Delta_{N+1}^{\pi_{\text{old}}}(s) \geq \Delta_N^{\pi_{\text{old}}}(s)$)**

We examine the difference between consecutive terms using the integral form:

$$\Delta_{N+1}^{\pi_{\text{old}}}(s) - \Delta_N^{\pi_{\text{old}}}(s) = \int_{-\infty}^\infty (F_X(x) - [F_X(x)]^{N+1})dx - \int_{-\infty}^\infty (F_X(x) - [F_X(x)]^N)dx$$

$$= \int_{-\infty}^\infty ([F_X(x)]^N - [F_X(x)]^{N+1})dx$$

$$= \int_{-\infty}^\infty [F_X(x)]^N (1 - F_X(x))dx.$$

Since $F_X(x) \in [0, 1]$, both terms in the integrand, $[F_X(x)]^N$ and $(1 - F_X(x))$, are non-negative. Their product is therefore non-negative. The integral of this non-negative function is non-negative, which implies $\Delta_{N+1}^{\pi_{\text{old}}}(s) - \Delta_N^{\pi_{\text{old}}}(s) \geq 0$.

**3. Proof of special case ($\Delta_1^{\pi_{\text{old}}}(s) = 0$)**

Substituting $N = 1$ into the integral identity yields:

$$\Delta_1^{\pi_{\text{old}}}(s) = \int_{-\infty}^{\infty} (F_X(x) - [F_X(x)]^1)dx = \int_{-\infty}^{\infty} (F_X(x) - F_X(x))dx = \int_{-\infty}^{\infty} 0\, dx = 0.$$

This confirms the special case. □

## B  Supplementary results

### B.1  Numerical results of ablation study

Table 4: Ablation on the impact of IVC.

| Task | MVP ($\lambda = 0.0$) | MVP ($\lambda = 0.5$) | MVP ($\lambda = 1.0$) |
|------|------|------|------|
| Cube-triple-task3 | $0.65 \pm 0.05$ | $0.70 \pm 0.14$ | $\mathbf{0.71 \pm 0.06}$ |
| Cube-triple-task4 | $0.30 \pm 0.21$ | $0.44 \pm 0.08$ | $\mathbf{0.52 \pm 0.11}$ |

Table 5: Comparison with one-step variants of the aforementioned baselines.

| Task | FQL-Onestep | BFN-Onestep | QC-Onestep | MVP (ours) |
|------|------|------|------|------|
| Cube-triple-task3 | $0.00 \pm 0.01$ | $0.00 \pm 0.00$ | $0.02 \pm 0.03$ | $\mathbf{0.71 \pm 0.06}$ |
| Cube-triple-task4 | $0.00 \pm 0.00$ | $0.00 \pm 0.00$ | $0.01 \pm 0.01$ | $\mathbf{0.52 \pm 0.11}$ |

## C  Environments Description

**Robomimic benchmark.**   We use three challenging tasks from the robomimic domain (Mandlekar et al., 2021). We use the multi-human datasets that were collected by six human operators. Each dataset consists of 50 trajectories provided by each operator, for a total of 300 successful trajectories. The operators were varied in proficiency – there were 2 "worse" operators, 2 "okay" operators, and 2 "better" operators, resulting in diverse, mixed quality datasets. The three tasks are as described below.

**(1) lift**: requires the robot arm to pick a small cube. This is the simplest task of the benchmark.

**(2) can**: requires the robot arm to pick up a coke can and place in a smaller container bin.

**(3) square**: requires the robot arm to pick a square nut and place it on a rod. The nut is slightly bigger than the rod and requires the arm to move precisely to complete the task successfully.

All of the three robomimic tasks use binary task completion rewards where the agent receives $-1$ reward when the task is not completed and $0$ reward when the task is completed.

**OGBench** `cube-double/triple`: These three domains contain 2/3 cubes respectively. The tasks in the two domains all involve moving the cubes to their desired locations. The reward is $-n_{\text{wrong}}$ where $n_{\text{wrong}}$ is the number of the cubes that are at the wrong position. The episode terminates when all cubes are at the correct position (reward is 0).

Below we highlight three representative tasks from each environment:

**(4) Cube-double-task2 (move)**: Two cubes in different colors are initialized at $(0.35, -0.1, 0.02)$ and $(0.50, -0.1, 0.02)$, and the goal is to move them to $(0.35, 0.1, 0.02)$ and $(0.50, 0.1, 0.02)$.

**(5) Cube-double-task3 (move)**: The initial cube positions are $(0.35, 0.0, 0.02)$ and $(0.50, 0.0, 0.02)$, and they must be placed at $(0.425, -0.2, 0.02)$ and $(0.425, 0.2, 0.02)$.

**(6) Cube-double-task4 (swap)**: Two cubes start from $(0.425, -0.1, 0.02)$ and $(0.425, 0.1, 0.02)$, and the objective is to swap their positions to $(0.425, 0.1, 0.02)$ and $(0.425, -0.1, 0.02)$.

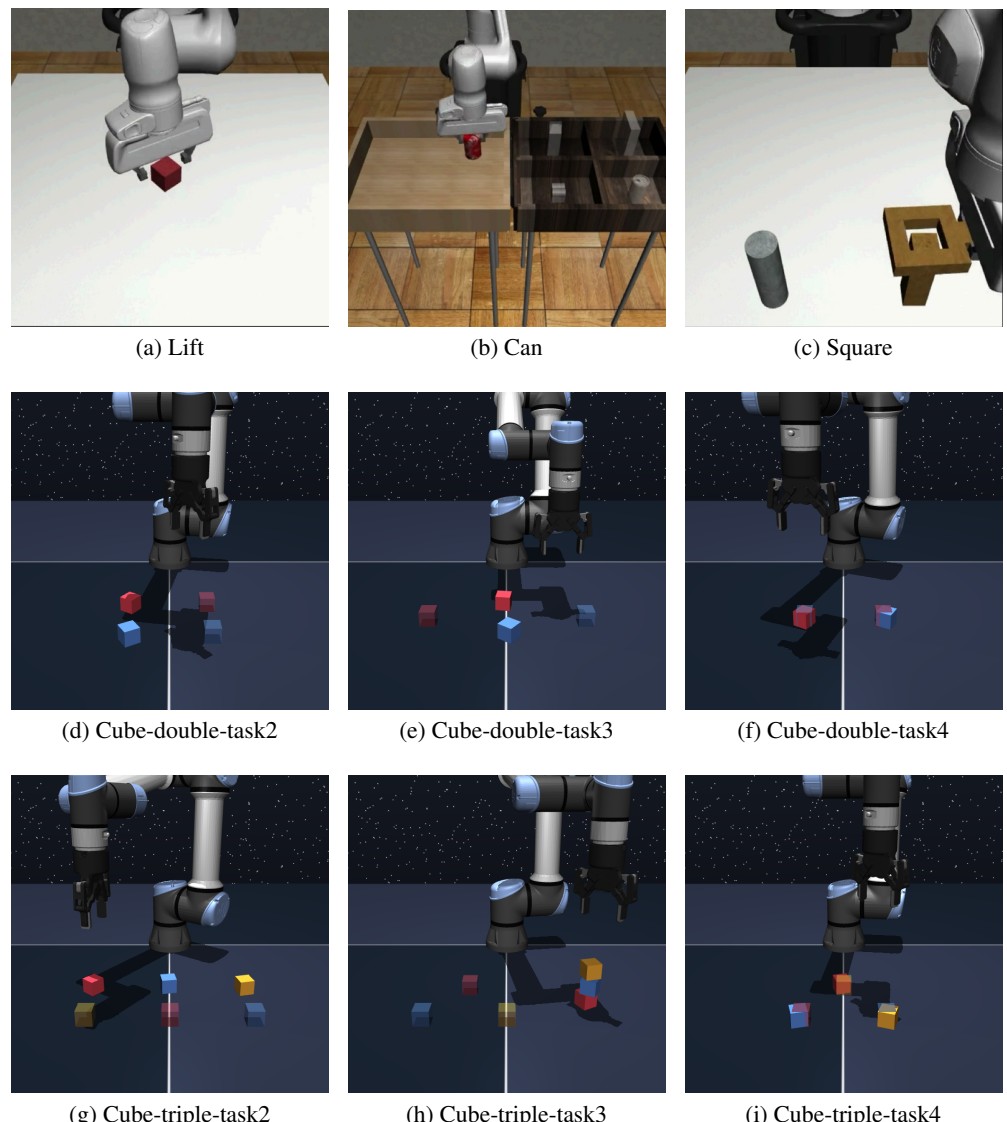

(a) Lift      (b) Can      (c) Square

(d) Cube-double-task2      (e) Cube-double-task3      (f) Cube-double-task4

(g) Cube-triple-task2      (h) Cube-triple-task3      (i) Cube-triple-task4

Figure 6: **Snapshots of the 9 challenging long-horizon, sparse-reward manipulation tasks.**

**(7) Cube-triple-task2 (move)**: Three cubes are initialized at $(0.35, -0.2, 0.02)$, $(0.35, 0.0, 0.02)$, and $(0.35, 0.2, 0.02)$, with goals at $(0.50, 0.0, 0.02)$, $(0.50, 0.2, 0.02)$, and $(0.50, -0.2, 0.02)$.

**(8) Cube-triple-task3 (stack)**: A stacked tower of three cubes is initialized at $(0.425, 0.2, 0.02)$, $(0.425, 0.2, 0.06)$, and $(0.425, 0.2, 0.10)$, and the robot must relocate them to $(0.35, -0.1, 0.02)$, $(0.50, -0.2, 0.02)$, and $(0.50, 0.0, 0.02)$, respectively.

**(9) Cube-triple-task4 (swap)**: Three cubes are initialized at $(0.35, 0.0, 0.02)$, $(0.50, -0.1, 0.02)$, and $(0.50, 0.1, 0.02)$, and they must be cyclically rearranged to $(0.50, -0.1, 0.02)$, $(0.50, 0.1, 0.02)$, and $(0.35, 0.0, 0.02)$.

These tasks highlight increasingly complex spatial reasoning requirements, ranging from coordinated multi-cube relocation to disassembly of stacked structures and cyclic rearrangement.

# D  VISUALIZATIONS

This section provides supplementary visual material to demonstrate our model's performance. The included visualizations of successful episodes highlight our policy's ability to generate precise and robust trajectories, showcasing its effectiveness in handling the complexities of **long-horizon reasoning** and **sparse rewards** inherent in these robotic manipulation tasks.

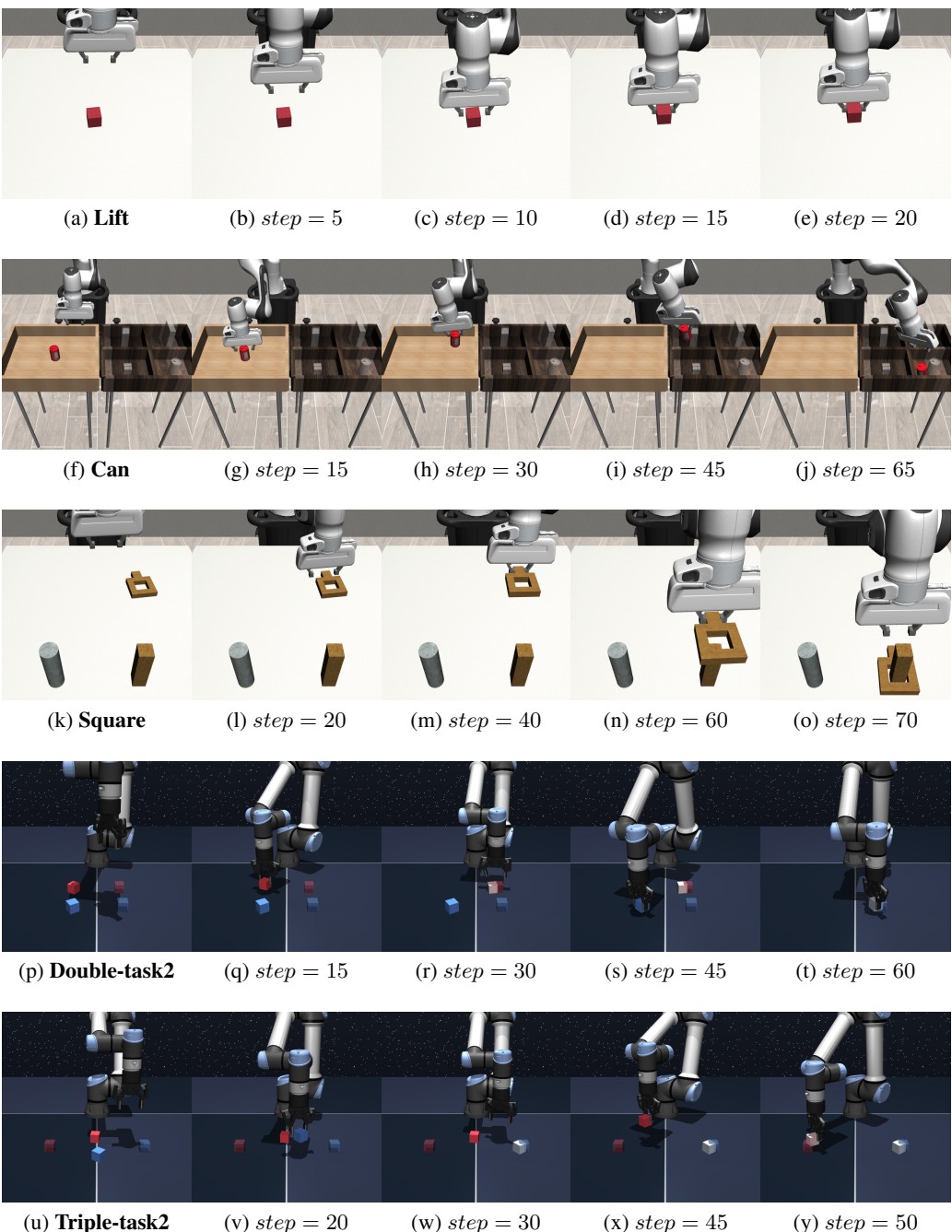

Figure 7: **Visualizations of typical success episodes: Robomimic-lift, Robomimic-can, Robomimic-square, Cube-double-task2, and Cube-double-task3.**

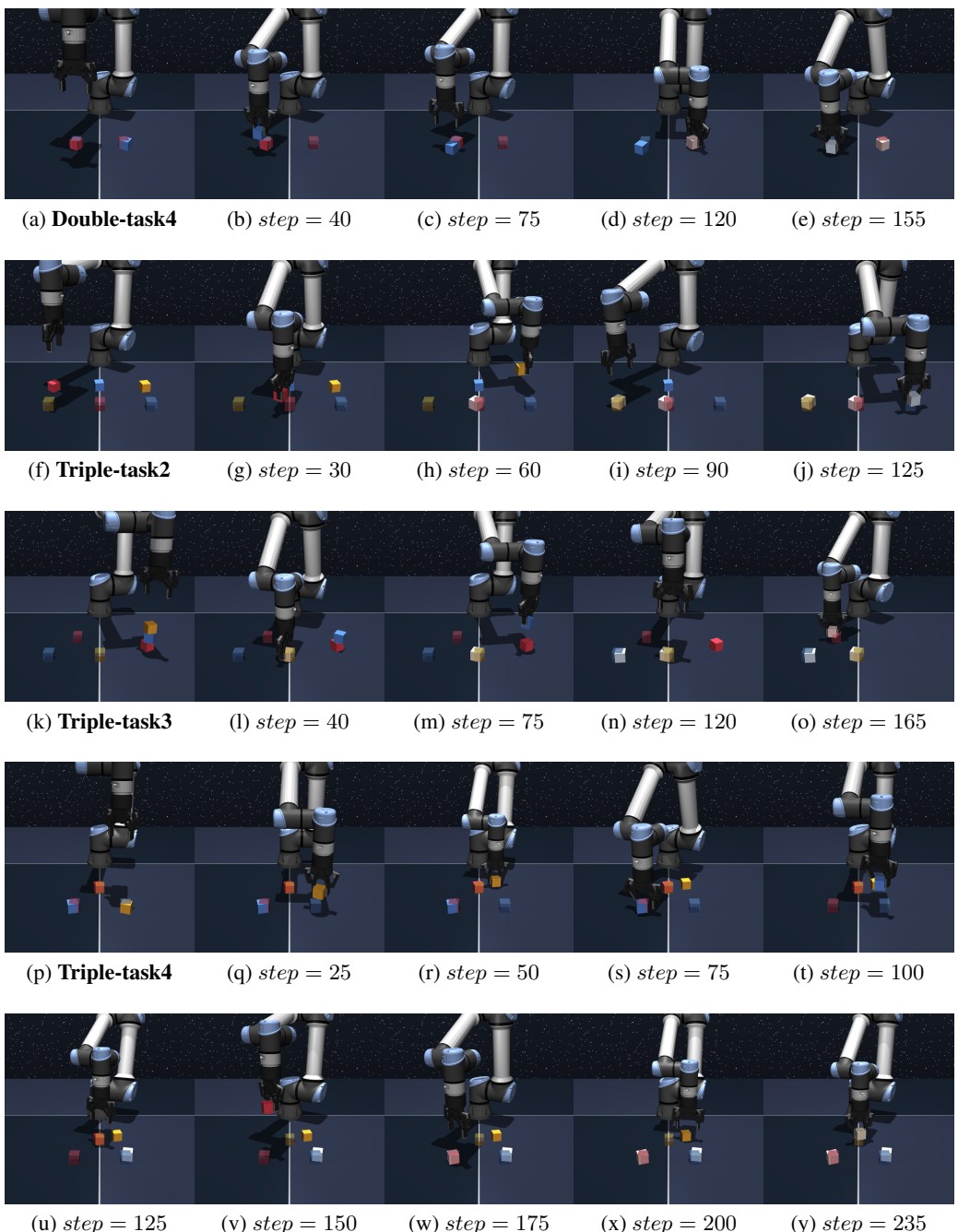

(a) **Double-task4**   (b) $step = 40$   (c) $step = 75$   (d) $step = 120$   (e) $step = 155$

(f) **Triple-task2**   (g) $step = 30$   (h) $step = 60$   (i) $step = 90$   (j) $step = 125$

(k) **Triple-task3**   (l) $step = 40$   (m) $step = 75$   (n) $step = 120$   (o) $step = 165$

(p) **Triple-task4**   (q) $step = 25$   (r) $step = 50$   (s) $step = 75$   (t) $step = 100$

(u) $step = 125$   (v) $step = 150$   (w) $step = 175$   (x) $step = 200$   (y) $step = 235$

Figure 8: **Visualizations of typical success episodes: Cube-double-task4, Cube-triple-task2, Cube-triple-task3, and Cube-triple-task4.**

# E    REPRODUCIBILITY STATEMENT

The hyperparameters of all algorithms are shown in Table 6.

| Parameter | Value |
|---|:---:|
| ***Shared*** | |
| Batch size | 256 |
| Discount factor ($\gamma$) | 0.99 |
| Optimizer | Adam |
| Learning rate | $3 \times 10^{-4}$ |
| Target network update rate ($\tau$) | $5 \times 10^{-3}$ |
| UTD Ratio | 1 |
| Evaluation interval | 5000 |
| Number of evaluation episodes | 50 |
| Number of offline training steps | $1 \times 10^6$ (1M) |
| Number of online training steps | $1 \times 10^6$ (1M) |
| Number of flow steps ($T$) | 10 |
| Policy network width | 512 |
| Policy network depth | 4 hidden layers |
| Policy activation function | GELU |
| Policy layer normalization | False |
| Value network width | 512 |
| Value network depth | 4 hidden layers |
| Value activation function | GELU |
| Value layer normalization | True |
| Value ensemble size ($K$) | 2 |
| Value ensemble operator | MEAN |
| ***FQL*** | |
| Flow step | 10 |
| BC weight ($\alpha$) | 10000 for `lift`, `can` and `square` 
 300 for `cube-double-*` and `cube-triple-*` |
| ***QC*** | |
| Chunking horizon length | 5 |
| Flow step | 10 |
| Number of best-of-$N$ | 16 for `lift`, `can` and `square` 
 32 for `cube-double-*` and `cube-triple-*` |
| ***MVP-IVC (ours)*** | |
| IVC ratio ($\lambda$) | 1.0 |
| Number of best-of-$N$ | 16 for `lift`, `can` and `square` 
 32 for `cube-double-*` and `cube-triple-*` |

Table 6: **Detailed hyperparameters.**

## F    LLM Usage Disclosure

We used ChatGPT to refine the grammar and improve the clarity of the text. All LLM-generated suggestions were reviewed and edited by the authors, who take full responsibility for the final content.

