# OpenReview forum: "Mean Flow Policy with Instantaneous Velocity Constraint for One-step Action Generation"
_ICLR.cc/2026/Conference — ICLR 2026 Oral_

### Official Review · Reviewer_ac7s · 2025-10-31

**Soundness:** 4
**Presentation:** 4
**Contribution:** 3
**Rating:** 8
**Confidence:** 3

**Summary:**

This paper proposes builds on the use of generative modelling for policy learning by introducing a new policy learning framework with MeanFlow, a one-step alternative to popularly used flow matching based approaches. Additionally, to improve the accuracy of the policy learning, the authors propose a new constraint on the MeanFlow generation framework that ensures the consistency of the instantaneous velocity in modelling the velocity field of flow matching. This is proven both theoretically and experimentally. The policy is then trained in an offline-to-online reinforcement learning setting wherein the Q-values from the critic are used to select the action targets in a best-of-N manner to train the MeanFlow policy network.

**Strengths:**

The proposed approach shows a novel way of modelling policies using MeanFlow rather than a naive approach that just shows the use of a new generative modelling method to policy learning. The authors show how their instantaneous velocity constraint helps improve the policy learning. The results show a good improvement over previous approaches that use different combinations of flow matching and a best-of-N selection. The presentation clarity of the paper is easy to follow, and the contributions are highlighted well both in terms of the writing and in terms of experimentation.

**Weaknesses:**

Given the growing use of generative modelling in Imitation Learning settings, and since the authors explore the offline-to-online RL case, a comparison in a simpler behavioral cloning paradigm can help show the advantage of the proposed approach better, which in general can be useful for learning much more complex tasks.

**Questions:**

- How well does the approach perform with an image-based backbone?
- Since the authors train their model from offline data as well, does similar performance hold for Imitation Learning as well?

---

> ### Author Response · Authors · 2025-11-21
>
> We sincerely appreciate your detailed and constructive feedback. Below are our responses to your comments.
>
> ## W1&Q2 (Performance in Imitation Learning Settings)
> > Given the growing use of generative modelling in Imitation Learning settings, and since the authors explore the offline-to-online RL case, a comparison in a simpler behavioral cloning paradigm can help show the advantage of the proposed approach better, which in general can be useful for learning much more complex tasks.
>
> > Since the authors train their model from offline data as well, does similar performance hold for Imitation Learning as well?
>
> We appreciate the insightful suggestion to evaluate our method within the Imitation Learning paradigm. In response, we conducted a pure Behavioral Cloning (BC) experiment comparing our one-step mean flow policy against a standard multi-step flow matching policy baseline using the provided offline datasets.
>
> The results, shown in [Figure 6](https://anonymous.4open.science/r/ICLR2026_MFP_Rebuttal-2256/Figure6_results_bc.pdf) (Figure 14 in the revised paper) and the following Table, demonstrate that **our MFP matches the strong multi-step baseline in success rate while maintaining significantly faster inference (1 step vs. 20 steps)**. This confirms that the advantages of MFP are robust and extend effectively to the generative imitation learning setting.
>
> | Algorithm | Robomimic-Square      |
> |-----------|--------------------|
> | Flow (1 step) | 0.15 ± 0.03        |
> | Flow (5 steps) | 0.46 ± 0.17        |
> | Flow (20 steps) | 0.48 ± 0.06        |
> | **MFP (ours)** | **0.48 ± 0.09**    |
>
> Additionally, we conducted a verification experiment on the multi-modality of our MFP using the Push-T task. As illustrated in [Figure 7](https://anonymous.4open.science/r/ICLR2026_MFP_Rebuttal-2256/Figure7_pushT.pdf) (Figure 15 in the revised paper), **MFP successfully captures two optimal action modes corresponding to different pushing strategies (left and right), validating its capability to model complex, multi-modal behaviors**.
>
> ## Q1 (Evaluation with Image-Based Backbone)
> > How well does the approach perform with an image-based backbone?
>
> We have validated our method on **visual-based manipulation tasks (Puzzle-3x3 and Puzzle-4x4)** using a ResNet-based encoder to process high-dimensional pixel inputs. As shown in [Figure 3](https://anonymous.4open.science/r/ICLR2026_MFP_Rebuttal-2256/Figure3_results_visual.pdf) (Figure 11 in the revised paper), MFP successfully learns effective policies from high-dimensional pixel inputs.
>
> - **On Visual-puzzle-3x3**: Both MFP and the strong baseline QC achieve perfect success rates, confirming that MFP works robustly with image backbones on standard tasks.
>
> - **On Visual-puzzle-4x4**: This harder task exposes the performance gap. We observe relatively high variance for both algorithms, which reflects the inherent difficulty of learning long-horizon planning from pixels with sparse rewards. Despite this challenge, MFP achieves a higher average success rate (0.75 vs. 0.47). This indicates that with an image backbone, MFP is more capable of extracting effective features for spatial reasoning and reaching peak performance than the baseline.
>
> | Algorithm | Visual-puzzle-3x3       | Visual-puzzle-4x4      |
> |-----------|--------------------|---------------------|
> | QC | 1.0 ± 0.0        | 0.47 ± 0.33        |
> | **MFP (ours)** | **1.0 ± 0.0**    | **0.75 ± 0.15**    |
>
> ## Final thanks
> Thank you again for your time, effort, and professionalism. We hope our responses address your questions and concerns. We are happy to provide additional details if needed, and we look forward to further discussion.

---

### Official Review · Reviewer_dkbj · 2025-11-01

**Soundness:** 3
**Presentation:** 3
**Contribution:** 2
**Rating:** 4
**Confidence:** 3

**Summary:**

This paper presents Mean Flow Policy (MFP), a reinforcement learning approach that adapts the mean-flow generative modeling framework for efficient one-step action generation. Instead of learning the instantaneous velocity field as in standard flow matching, MFP models the mean velocity field, enabling direct sampling without multi-step integration. To address the underdetermined mean-flow ODE, the authors add an Instantaneous Velocity Constraint (IVC) enforcing equality between mean and instantaneous velocities at the boundary r=t. Combined with a best-of-N Q-guided action selection scheme, MFP achieves competitive or superior results to existing flow-based RL methods (FQL, BFN, QC) on Robomimic and OGBench, while significantly improving training and inference speed.

**Strengths:**

- Clear theoretical presentation with formal analysis of the mean-flow ODE’s non-uniqueness and the effect of the IVC boundary constraint.
- Empirical results show consistent efficiency gains over multi-step flow policies.
- The IVC ablation confirms its stabilizing effect during training.
- Writing and experimental setup are clear and reproducible.

**Weaknesses:**

- The mean-flow formulation itself is taken directly from prior generative modeling work (*Geng et al., 2025a*); this paper mainly applies it to RL.
- The proposed IVC is conceptually implied by the mean-flow definition when r\!\to\!t, so it should not be considered a new theoretical contribution.
- The best-of-N action-selection mechanism is standard in prior RL methods (EMaQ, BFN, FQL).
- The experimental scope is limited: only a small number of state-based manipulation tasks are used, with no visual-policy or diverse-domain evaluation. This constrains the empirical strength of the claims.

**Questions:**

The IVC condition appears to be mathematically implied by the mean-flow definition (Eq. 6) and the mean-flow loss (Eq. 9) as $r \rightarrow t$. The added IVC term seems to primarily emphasize the boundary condition rather than address a missing component of the mean-flow formulation. Could the authors clarify whether this term introduces any new theoretical insight beyond reinforcing an already implied constraint?

---

> ### Author Response · Authors · 2025-11-21
>
> We sincerely appreciate your detailed and constructive feedback. Below are our responses to your comments.
>
> ## W1 (Theoretical Novelty in Generative RL Context)
> > The mean-flow formulation itself is taken directly from prior generative modeling work (Geng et al., 2025a); this paper mainly applies it to RL.
>
> **Fully releasing the power of generative modeling in RL is non-trivial**, as RL introduces unique challenges not present in generative modeling for static data (e.g., images). Unlike image generation where the dataset is fixed and independent, the non-stationary nature of online RL (policy updates change the data distribution) requires the generative model to adapt dynamically and quickly while maintaining policy stability.
>
> To address this, we first provide a theoretical guarantee (Theorem 1) for the "generate-and-select" mechanism, ensuring the theoretical soundness of employing generative flow models as policies in RL. Additionally, we identify and resolve a fundamental theoretical deficiency of the mean-flow formulation in RL: its underdetermined nature. We formally prove that without an explicit boundary condition, the standard mean-flow objective allows for infinite solutions (Theorem 2). Our proposed IVC is theoretically derived as the necessary boundary condition to guarantee solution uniqueness (Theorem 3), transforming the method from an empirical heuristic (when directly applied to RL) into a well-posed policy class.
>
> ## W2&Q1 (Theoretical Necessity of Explicit IVC Constraint)
> > The proposed IVC is conceptually implied by the mean-flow definition when $r \to t$, so it should not be considered a new theoretical contribution.
> > The IVC condition appears to be mathematically implied by the mean-flow definition (Eq. 6) and the mean-flow loss (Eq. 9) as $r \to t$. The added IVC term seems to primarily emphasize the boundary condition rather than address a missing component of the mean-flow formulation. Could the authors clarify whether this term introduces any new theoretical insight beyond reinforcing an already implied constraint?
>
> We thank the reviewer for this insightful observation. We fully concur that the mean flow definition (Eq. 6) analytically implies the instantaneous velocity constraint in the limit $r \to t$. However, our key contribution with IVC is not to propose a new mathematical definition, but to address two critical gaps when applying the mean-flow formulation to RL: **optimization intractability of implicit constraints** and **theoretical ill-posedness of learning this definition via mean-flow loss (Eq. 9)**. Specifically, the critical role of IVC is established through the following three arguments:
>
> - **Implicit Constraint vs. Explicit Supervision.** While the mean flow loss (Eq. 9) theoretically governs the flow dynamics, relying only on it requires the network to implicitly satisfy the boundary condition by fitting dynamics across various time intervals. This "implicit learning" is numerically challenging. IVC translates this requirement into an explicit supervision signal. This decouples the learning of the boundary condition from the fitting of flow dynamics, stabilizing the optimization process.
>
> - **Theoretical Necessity (Theorem 2 & 3).** As proven in Theorem 2, the mean flow loss (Eq. 9) suffers from a multiplicity of solutions—meaning the optimization objective admits a family of incorrect solutions characterized by an arbitrary integration constant. Mathematically, minimizing Eq. 9 alone acts as solving a differential equation without a boundary condition. Therefore, IVC is not redundant; it serves as the necessary boundary condition (as formally established in Theorem 3) that eliminates this degree of freedom, transforming the optimization from an ill-posed problem into a well-posed one.
>
> - **Empirical Validation.** Our ablation study supports this analysis (Figure 4 and Table 4 in the revised paper). Without IVC, the model fails to learn the correct flow effectively, resulting in a significant performance drop (success rate falls from 0.52 to 0.30 on Cube-triple-task4). This confirms that explicit boundary supervision is a prerequisite for successful training.

---

> ### Author Response · Authors · 2025-11-21
>
> ## W3 (Distinction from Prior "Best-of-N" Methods)
>
> > The best-of-N action-selection mechanism is standard in prior RL methods (EMaQ, BFN, FQL).
>
> While the "Best-of-N" heuristic is known, our contribution lies in establishing the first theoretical justification for the "generate-and-select" paradigm as a policy improvement mechanism for generative flow policies in RL. This is fundamentally distinct from prior theoretical work (i.e., EMaQ [1]) in two key ways:
>
> First, EMaQ uses the best-of-N mechanism as a Bellman operator to define the target value for the Q-function (**Critic-only Q-learning architecture**). It provides no mechanism for learning or updating the policy itself. Our work defines best-of-N as a policy improvement mechanism for the actor (**Actor-Critic architecture**). The selected action $a^*$ serves as the explicit target for training our generative flow policy, thereby providing a clear target for policy learning.
>
> Second, EMaQ's theory is confined to the tabular setting. Our Theorem 1 provides the policy improvement guarantee for the 'generate-and-select' policy update mechanism in the general function approximation setting. It is the first to formally quantify the trade-off between the best-of-N advantage gain $\Delta_{N}^{\pi_{old}}$ and the critic and flow matching approximation errors ($\epsilon_Q, \epsilon_A$).
>
> Regarding practical implementation, the algorithms also differ: EMaQ is a value-based method that lacks an explicit actor update; BFN baseline in our experiments refers to an adaptation of the Best-of-N strategy (from EMaQ) applied to a flow-based Actor-Critic architecture, which serves as a strong empirical baseline; and FQL [2] relies on policy distillation and maximizing Q-values via backpropagation, without utilizing the best-of-N mechanism.
>
> [1] *Ghasemipour, et al. "Emaq: Expected-max q-learning operator for simple yet effective offline and online rl." International Conference on Machine Learning. PMLR, 2021*.
>
> [2] *Park, Seohong, et al. "Flow q-learning." arXiv preprint arXiv:2502.02538 (2025)*.
>
> ## W4 (Additional Evaluation on High-Dimensional and Visual Domains)
>
> > The experimental scope is limited: only a small number of state-based manipulation tasks are used, with no visual-policy or diverse-domain evaluation. This constrains the empirical strength of the claims.
>
> Thanks for your suggestion. To demonstrate the effectiveness of MFP across diverse domains and modalities, we have incorporated two distinct types of challenging benchmarks, as shown in [Figure 1](https://anonymous.4open.science/r/ICLR2026_MFP_Rebuttal-2256/Figure1_additional_env_intro.pdf) (Figure 9 in the revised paper).
>
> - **High-Dimensional Control (D4RL Adroit)**: To test complex control dynamics, we evaluated MFP on the **D4RL Adroit suite (Hammer and Door)**. These tasks represent high-dimensional dexterous manipulation challenges involving precise tool use and latch coordination. As shown in [Figure 2](https://anonymous.4open.science/r/ICLR2026_MFP_Rebuttal-2256/Figure2_results_d4rl.pdf) (Figure 10 in the revised paper) and the table below, MFP demonstrates superior capability in modeling complex action distributions. Specifically, MFP outperforms the strong baseline QC by a significant margin, achieving a 14-point increase in normalized score on average. This result strongly verifies MFP's scalability to high-DoF control problems.
>
> | Algorithm | D4RL-Door       | D4RL-Hammer      |
> |-----------|--------------------|---------------------|
> | QC | 51.8 ± 6.0        |    92.2 ± 4.5     |
> | **MFP (ours)** | **64.7 ± 4.7**    | **108.3 ± 5.2**    |
>
> - **Visual-Based Manipulation (Puzzle)**: To address the concern on visual policies, we conducted experiments on the **visual-based Puzzle-3x3 and Puzzle-4x4 tasks** shown in [Figure 3](https://anonymous.4open.science/r/ICLR2026_MFP_Rebuttal-2256/Figure3_results_visual.pdf) (Figure 11 in the revised paper), which require spatial reasoning directly from pixel inputs. The numerical results are summarized below. While Puzzle-3x3 is well solved by both methods, the significantly harder Puzzle-4x4 presents a greater challenge. We observe relatively higher variance for both algorithms on Puzzle-4x4, which reflects the inherent difficulty of learning long-horizon planning from pixels with sparse rewards. Despite this challenge, MFP achieves a higher average success rate (0.75 vs. 0.47). This indicates that MFP is more capable of exploring and reaching peak performance in complex visual domains compared to the baseline.
>
> | Algorithm | Visual-puzzle-3x3 | Visual-puzzle-4x4 |
> |-----------|--------------------|---------------------|
> | QC | 1.0 ± 0.0        | 0.47 ± 0.33        |
> | **MFP (ours)** | **1.0 ± 0.0**    | **0.75 ± 0.15**    |
>
> ## Final thanks
> Thank you again for your time, effort, and professionalism. We hope our responses address your questions and concerns. We are happy to provide additional details if needed, and we look forward to further discussion.

---

### Official Review · Reviewer_aBKZ · 2025-11-03

**Soundness:** 4
**Presentation:** 3
**Contribution:** 4
**Rating:** 8
**Confidence:** 4

**Summary:**

This paper introduces the Mean Flow Policy (MFP), a novel generative policy for reinforcement learning that is both highly expressive and computationally efficient.

The core idea is to represent the policy using MeanFlow [1]. As MeanFlow is trained via supervised learning, it is incompatible with standard reinforcement learning frameworks. To bridge this gap, the proposed method samples N actions from the policy, selects the action with the highest Q-value, and uses this optimal action as a supervised target to train the MeanFlow policy.

Furthermore, to ensure stable and accurate learning, an Instantaneous Velocity Constraint (IVC) is introduced. This constraint imposes a necessary boundary condition on the underlying ordinary differential equation, which stabilizes the training process and enhances model fidelity.

Empirically, MFP achieves state-of-the-art success rates on several challenging robotic manipulation benchmarks (Robomimic and OGBench), while demonstrating significant computational speedups over existing multi-step, flow-based baselines.

[1] Geng, Zhengyang, et al. "Mean flows for one-step generative modeling." arXiv preprint arXiv:2505.13447 (2025).

**Strengths:**

- It has always been a question for me how diffusion / flow-matching policies can be used in RL. The “generate-and-select” mechanism used in this paper seems simple and straightforward, yet the authors demonstrate that it works remarkably well. I really appreciate this idea.

- By incorporating recent advances from generative AI (specifically, MeanFlow), the method enables fast online RL training while preserving the generative model’s ability to represent complex, multimodal action distributions.

- The approach achieves the highest or second-highest success rates across nine challenging robotic manipulation benchmarks, showcasing its effectiveness in solving complex, long-horizon tasks.

- The authors provide rigorous mathematical proofs to justify their framework. They show that the implicit value constraint (IVC) ensures a unique solution for the mean flow identity and that their overall algorithm guarantees consistent policy improvement.

**Weaknesses:**

- The paper emphasizes its one-step inference speed but understates the associated training cost. The core training loss (Eq. 9) necessitates a Jacobian-vector product (JVP) to compute the $\frac{d}{dt} \mathbf{u}_\theta$ term. This operation is often incompatible with optimized attention implementations like FlashAttention, potentially limiting the method's training efficiency and scalability.

- The paper repeatedly claims suitability for "real-time control systems" and "real-time deployment," yet these claims lack empirical validation on physical robotic hardware, as all experiments are conducted in simulation.

- The theoretical proofs for policy improvement (Theorem 1) and solution uniqueness (Theorem 3) are elegant but rely on strong assumptions that are often violated in practice. For instance, Theorem 1 assumes bounded Q-function and mean flow matching errors (epsilon_Q, epsilon_A). In deep RL, these errors can be large and unstable. Similarly, the proof for IVC's effectiveness assumes the mean flow loss (L_MF) is perfectly minimized, which is never the case with non-convex optimization. While the theory provides intuition, its practical guarantees are much weaker than presented.

**Questions:**

- In the MeanFlow paper [1], the authors train the MeanFlow model using the MeanFlow objective for only 25% of the training time, while employing the Flow Matching objective (where r=t) for the remaining 75%. Similarly, FACM [2] incorporates the original Flow Matching loss into its training objective. How does the Instantaneous Velocity Constraint (IVC) proposed in this paper differ from them?


[1] Geng, Zhengyang, et al. "Mean flows for one-step generative modeling." arXiv preprint arXiv:2505.13447 (2025).

[2] Peng, Yansong, et al. "Flow-anchored consistency models." arXiv preprint arXiv:2507.03738 (2025).

---

> ### Author Response · Authors · 2025-11-21
>
> We sincerely appreciate your detailed and constructive feedback. Below are our responses to your comments.
>
> ## W1 (Training Efficiency and FlashAttention Compatibility)
> > The paper emphasizes its one-step inference speed but understates the associated training cost. The core training loss (Eq. 9) necessitates a Jacobian-vector product (JVP) to compute the term. This operation is often incompatible with optimized attention implementations like FlashAttention, potentially limiting the method's training efficiency and scalability.
>
> We clarify that Figure 1 in the original paper already quantifies the training efficiency, explicitly capturing the JVP overhead. The figure compares the online training speed (measured in iterations per second) of our MFP against baselines.
>
> Regarding the concern about FlashAttention compatibility, we clarify that this is no longer a limitation. **Efficient JVP-supported FlashAttention kernels have been successfully implemented and verified by the open-source community** (e.g., see the implementation in [1] and discussions in [2]). These developments confirm that the JVP operation in our loss function is fully compatible with optimized attention mechanisms, ensuring the method's scalability.
>
> [1] https://github.com/amorehead/jvp_flash_attention
>
> [2] https://github.com/Dao-AILab/flash-attention/issues/1672
>
> ## W2 (Validation of Real-Time Inference Feasibility)
> > The paper repeatedly claims suitability for "real-time control systems" and "real-time deployment," yet these claims lack empirical validation on physical robotic hardware, as all experiments are conducted in simulation.
>
> While our experiments are simulation-based, we validated the computational feasibility for real-time control loops by benchmarking inference on a **CPU-only environment** (AMD Threadripper). The deployed policy is a standard MLP with a single function evaluation. It achieves ~10ms latency (Table 3 in the revised paper), comfortably satisfying the strict timing requirements of high-frequency (e.g., 50-100Hz) [3] physical control loops. This CPU-only latency is directly transferable to physical robotic hardware, as it reflects the core computational overhead of the policy (a standard MLP with single function evaluation) without simulation-specific overhead. This confirms that MFP eliminates the computational bottleneck of generative control, making it highly compatible with existing real-time hardware stacks without the need for specialized accelerators.
>
> [3] *Hwangbo, Jemin, et al. "Learning agile and dynamic motor skills for legged robots." Science Robotics 4.26 (2019): eaau5872.*
>
> ## W3 (Justification of Theoretical Assumptions and IVC Necessity)
> > The theoretical proofs for policy improvement (Theorem 1) and solution uniqueness (Theorem 3) are elegant but rely on strong assumptions that are often violated in practice. For instance, Theorem 1 assumes bounded Q-function and mean flow matching errors (epsilon_Q, epsilon_A). In deep RL, these errors can be large and unstable. Similarly, the proof for IVC's effectiveness assumes the mean flow loss (L_MF) is perfectly minimized, which is never the case with non-convex optimization. While the theory provides intuition, its practical guarantees are much weaker than presented.
>
> We acknowledge that our theorems rely on simplifying assumptions, but such analysis are standard and common for mathematical tractability in RL literature. For instance, assumptions on bounded Q-function fitting errors and distribution matching errors are widely adopted to establish theoretical foundations, as seen in works like LOOP (Theorem 1) [4] and PRDC (Theorem 3.7) [5].
>
> Regarding the minimization of $L_{MF}$, our theoretical argument does not assume perfect optimization in practice. Rather, it demonstrates that **even under ideal optimization, the standard objective remains underdetermined without a boundary condition.** Thus, IVC is derived not as a heuristic, but as the mathematically required operator to resolve this ambiguity. Our empirical results validate the robustness of the method despite some practical deviations from these assumptions.
>
> [4] *Sikchi, Harshit, Wenxuan Zhou, and David Held. "Learning off-policy with online planning." Conference on Robot Learning. PMLR, 2022.*
>
> [5] *Ran, Yuhang, et al. "Policy regularization with dataset constraint for offline reinforcement learning." International conference on machine learning. PMLR, 2023.*

---

> ### Author Response · Authors · 2025-11-21
>
> ## Q1 (Differentiation from MeanFlow and FACM)
> > In the MeanFlow paper, the authors train the MeanFlow model using the MeanFlow objective for only 25% of the training time, while employing the Flow Matching objective (where r=t) for the remaining 75%. Similarly, FACM incorporates the original Flow Matching loss into its training objective. How does the Instantaneous Velocity Constraint (IVC) proposed in this paper differ from them?
>
> We thank the reviewer for highlighting the connections to MeanFlow [6] and the concurrent work FACM [7]. While these methods all incorporate flow matching objectives, our IVC differs fundamentally from them in both implementation mechanism and theoretical role.
>
> - **Comparison with MeanFlow**: The distinction lies in the training methodology. MeanFlow adopts a heuristic stochastic mixing strategy that applies the Mean Flow loss and the Flow Matching loss to disjoint subsets of samples according to a mixing ratio. This approach depends on the neural network's implicit generalization, which we find insufficient for the non-stationary data distributions inherent to RL. In contrast, we implement IVC as a simultaneously enforced constraint on every sample. Grounded in the theoretical resolution of solution multiplicity, this design explicitly eliminates the integration drift $C/(r-t)$ at every optimization step, yielding superior stability. Our ablation studies in [Figure 5](https://anonymous.4open.science/r/ICLR2026_MFP_Rebuttal-2256/Figure5_ablation_on_mix_ratio.pdf) (Figure 13 in the revised paper) confirm that our IVC loss yields consistently higher success rates and stability, especially in the harder Task 4 where the gap widens (0.30 vs 0.52).
>
> | Algorithm | Cube-triple-task3       | Cube-triple-task4      |
> |-----------|--------------------|---------------------|
> | MFP (25\%) | 0.68 ± 0.08        | 0.31 ± 0.07        |
> | MFP (50\%) | 0.69 ± 0.07        | 0.30 ± 0.08        |
> | MFP (75\%) | 0.66 ± 0.03        | 0.25 ± 0.07        |
> | **MFP+IVC (ours)** | **0.71 ± 0.08**    | **0.52 ± 0.11**    |
>
> - **Comparison with FACM**: The methods address distinct mathematical problems. FACM introduces the Flow Matching objective primarily as an "anchor" to mitigate the optimization instability inherent to Consistency Models (CM). Specifically, they identify that the CM objective is "ungrounded", causing the self-referential derivative estimation (Eq. 7 in FACM) to become noisy and erratic, leading to training collapse. Our work, however, focuses on the Mean Flow formulation and addresses the problem of mathematical ill-posedness. As we prove in Theorem 2, the standard Mean Flow objective admits infinite solutions without a boundary condition. IVC is formally derived as the necessary condition to lock the unique correct solution. Therefore, while FACM uses flow matching to stabilize the learning process, we use it to mathematically define the unique solution of the differential equation itself.
>
> [6] *Geng, Zhengyang, et al. "Mean flows for one-step generative modeling." arXiv preprint arXiv:2505.13447 (2025).*
>
> [7] *Peng, Yansong, et al. "Flow-anchored consistency models." arXiv preprint arXiv:2507.03738 (2025).*
>
> ## Final thanks
> Thank you again for your time, effort, and professionalism. We hope our responses address your questions and concerns. We are happy to provide additional details if needed, and we look forward to further discussion.

---

> > ### Comment · Reviewer_aBKZ · 2025-11-27
> >
> > Thank you for the detailed response! I still recommend acceptance.

---

### Official Review · Reviewer_Ruko · 2025-11-03

**Soundness:** 4
**Presentation:** 3
**Contribution:** 3
**Rating:** 8
**Confidence:** 3

**Summary:**

This paper introduces the Mean Flow Policy (MFP) which is an improvement for flow-matching based policies using the idea of mean flows. The key insight is that in traditional flow policies, one has to integrate the instantaneous velocity from 0 to 1 to transform from the standard normal distribution to the output distribution which selects the action. Instead of learning instantaneous velocities (v(t)) and then performing numerical integration, this paper shows that you can learn a mean-flow (u(t,r)) which is the velocity from t to r. This allows one to get the target distribution in one step as opposed to the several needed in typical flow matching papers. To ensure accurate learning of this mean velocity, the authors propose an Instantaneous Velocity Constraint (IVC), which serves as a boundary condition to eliminate solution ambiguity in the mean flow ordinary differential equation. Theoretically, the paper proves that IVC ensures a unique and accurate solution and that MFP guarantees policy improvement under bounded fitting errors. Empirically, MFP achieves state-of-the-art success rates but provides training/inference speedups on two standard simulation benchmarks.

**Strengths:**

This is a strong paper that introduces a new formulation for flow matching, which is prevalent tool in robot learning. A major pain point for flow-matching or diffusion style approaches is the slow inference times. We need several denoising steps or numerical integration, that greatly slows down inference and the rate at which such policies can be applied. Unlike "hacks", this paper presents a clean reformulation using the ideas of mean-flow to reduce the inference time. The theoretical analysis is strong. I expect this to be the starting point for several follow-up ideas that can greatly improve the empirical performance as well.

**Weaknesses:**

An unfortunate (IMO) weakness of the paper is the limited empirical results. It uses only two simulation environments and the improvements are minimal over the baselines. In most cases, it matches or only slightly exceeds the baseline. In a way that's not surprising to me. The claim isn't that this is a better method (in terms of higher success rates) than baselines but that it is a faster method (during inference). However, the inference time comparisons are only in the appendix with some short description in the main body. I would hav liked to see more discussion on that aspect.

Another issue is that the two simulation seem "saturated" in that the baselines already perform pretty well. It would have been nice to look at more complex benchmarks.

It would have been especially nice to see how fixing the inference time (for example) results in better/worse performance for the several baselines. For diffusion style methods, it is not uncommon to perform comparisons with varying the number of denoising steps (which controls for the inference time). An equivalent study here would have really benefitted the paper.

**Questions:**

1. The paper’s main claim is that Mean-Flow Policy achieves faster inference than standard flow or diffusion policies. However, the runtime comparisons are only briefly mentioned in the appendix. Could you provide more details on the experimental setup for these timing results e.g., hardware used, batch size, and number of integration or denoising steps for baselines — and consider moving a quantitative runtime table into the main text?

2. The reported environments (Robomimic and OGBench) seem relatively saturated, with strong baseline performance. Have you considered evaluating on more complex or long-horizon benchmarks to better stress-test the method’s benefits? If not, could you explain the rationale behind choosing these two domains?

---

> ### Author Response · Authors · 2025-11-21
>
> We sincerely appreciate your detailed and constructive feedback. Below are our responses to your comments.
>
> ## W1&Q2 (Expanded Evaluation on High-Dimensional and Visual Tasks)
> > An unfortunate (IMO) weakness of the paper is the limited empirical results. It uses only two simulation environments and the improvements are minimal over the baselines. In most cases, it matches or only slightly exceeds the baseline. In a way that's not surprising to me. Another issue is that the two simulation seem "saturated" in that the baselines already perform pretty well. It would have been nice to look at more complex benchmarks.
> The reported environments (Robomimic and OGBench) seem relatively saturated, with strong baseline performance. Have you considered evaluating on more complex or long-horizon benchmarks to better stress-test the method’s benefits? If not, could you explain the rationale behind choosing these two domains?
>
> To further stress-test our method on higher-complexity domains, we have expanded our evaluation to include the high-dimensional **D4RL Adroit tasks (Hammer and Door)** to test complex control dynamics, and the **visual-based manipulation tasks (Puzzle-3x3 and Puzzle-4x4)** to evaluate performance under varying modalities, as shown in [Figure 1](https://anonymous.4open.science/r/ICLR2026_MFP_Rebuttal-2256/Figure1_additional_env_intro.pdf)  (Figure 9 in the revised paper).
>
> The results are shown in [Figure 2](https://anonymous.4open.science/r/ICLR2026_MFP_Rebuttal-2256/Figure2_results_d4rl.pdf) (Figure 10 in the revised paper) and [Figure 3](https://anonymous.4open.science/r/ICLR2026_MFP_Rebuttal-2256/Figure3_results_visual.pdf) (Figure 11 in the revised paper), with numerical results listed below.
>
> - **High-Dimensional Control (D4RL Adroit)**: On the Adroit benchmarks, MFP outperforms the strong baseline QC by a significant margin, achieving a 14-point increase in normalized score on average. This highlights MFP's superior capability in modeling complex action distributions in high-dimensional spaces.
>
> | Algorithm | D4RL-Door       | D4RL-Hammer      |
> |-----------|--------------------|---------------------|
> | QC | 51.8 ± 6.0        |   92.2 ± 4.5      |
> | **MFP (ours)** | **64.7 ± 4.7**    | **108.3 ± 5.2**    |
>
> - **Visual-Based Manipulation (Puzzle)**: In the visual domain, both methods successfully solve the Visual-puzzle-3x3 task. For the significantly harder Visual-puzzle-4x4 task, we observe relatively higher variance for both algorithms, which reflects the inherent challenge of learning long-horizon spatial reasoning from pixels with sparse rewards. Despite this difficulty, MFP achieves a higher average success rate (0.75 vs. 0.47) and demonstrates superior convergence and stability compared to the baseline.
>
> | Algorithm | Visual-puzzle-3x3       | Visual-puzzle-4x4      |
> |-----------|--------------------|---------------------|
> | QC | 1.0 ± 0.0        | 0.47 ± 0.33        |
> | **MFP (ours)** | **1.0 ± 0.0**    | **0.75 ± 0.15**    |
>
> Our initial focus on Robomimic and OGBench was driven by their status as standard benchmarks in recent flow-based RL literature, which ensures a fair comparison with state-of-the-art methods. Their well-established evaluation protocols and public datasets also ensure the reliability of our initial findings. By supplementing them with the more challenging D4RL Adroit and visual tasks, we provide a comprehensive assessment of our method’s performance across different complexity levels.

---

> ### Author Response · Authors · 2025-11-21
>
> ## W2 (Ablation Study on Baseline Inference Steps)
> > It would have been especially nice to see how fixing the inference time (for example) results in better/worse performance for the several baselines. For diffusion style methods, it is not uncommon to perform comparisons with varying the number of denoising steps (which controls for the inference time). An equivalent study here would have really benefited the paper.
>
> We highlight that MFP is structurally designed for one-step generation (NFE=1), theoretically achieving the fastest possible inference speed as it requires minimal computational overhead. In contrast, the diffusion baselines we consider typically rely on multi-step refinement to obtain high-quality solutions, and their inference time is directly tied to the number of refinement steps.
>
> Following your recommendation, we evaluated the baselines with varying inference steps to demonstrate the trade-off between efficiency and performance. As detailed in the newly supplemented [Figure 4](https://anonymous.4open.science/r/ICLR2026_MFP_Rebuttal-2256/Figure4_ablation_on_flow_steps.pdf) (Figure 12 in the revised paper), reducing the number of inference steps for the baselines results in a catastrophic performance drop. This confirms that standard flow policies collapse under low NFE budgets, whereas MFP retains SOTA performance.
>
> | Algorithm | Cube-triple-task3       | Cube-triple-task4      |
> |-----------|--------------------|---------------------|
> | QC (1 step) | 0.02 ± 0.02        | 0.01 ± 0.01        |
> | QC (5 steps) | 0.33 ± 0.40        | 0.08 ± 0.04        |
> | QC (20 steps) | 0.69 ± 0.07        | 0.43 ± 0.10        |
> | **MFP (ours)** | **0.71 ± 0.08**    | **0.52 ± 0.11**    |
>
> ## Q1 (Detailed Runtime Analysis and Hardware Setup)
> > The paper’s main claim is that Mean-Flow Policy achieves faster inference than standard flow or diffusion policies. However, the runtime comparisons are only briefly mentioned in the appendix. Could you provide more details on the experimental setup for these timing results e.g., hardware used, batch size, and number of integration or denoising steps for baselines — and consider moving a quantitative runtime table into the main text?
>
> Thank you for highlighting this critical point—we fully agree that detailed and transparent runtime comparisons are essential to supporting our core claim.
>
> We have revised Section 4.2 to detail the exact experimental setup used for the **inference latency analysis**.
> - **Inference hardware setup:** To evaluate inference efficiency in a general-purpose computing environment without hardware acceleration, inference latency was measured on a **consumer-grade CPU-only** platform (AMD Ryzen Threadripper 3960X) without JIT compilation.
> - **Inference benchmarking protocol:** We standardized the batch size to $B=1$ to replicate online control conditions. Baselines (BFN, QC) required 10 integration steps (NFE=10), whereas MFP and FQL operate with a fixed single step (NFE=1). Each method’s inference latency was measured over 1000 independent trials to alleviate random fluctuations.
>
> Accordingly, the quantitative runtime analysis (previously Table 4 in Appendix B.2) has been promoted to Section 4.2 in main text.
>
> ## Final thanks
> Thank you again for your time, effort, and professionalism. We hope our responses address your questions and concerns. We are happy to provide additional details if needed, and we look forward to further discussion.

---

### Author Response · Authors · 2025-12-03

Dear Area Chair and Reviewers,

Thank you for your engagement in reviewing our paper and for providing the valuable feedback. Below is a summary of our rebuttal.

We are encouraged that most of reviewers recommended acceptance with initial scores of 8 (**`Ruko`**), 8 (**`ac7s`**), 8 (**`aBKZ `**), 4 (**`dkbj`**). Reviewer **`aBKZ`** explicitly reaffirmed the acceptance score during the rebuttal. To address all reviewers’ concerns comprehensively, we implemented targeted actions:

# 1. Theoretical Clarifications (**`dkbj`**)
For concerns regarding the redundancy of the *Instantaneous Velocity Constraint (IVC)* and the novelty of the *Best-of-N* mechanism, we have clarified the following:

- **Necessity of IVC:** We proved that standard MeanFlow identity is ill-posed without a boundary condition in Theorem 2. IVC is derived as the necessary condition for uniqueness in Theorem 3 rather than a heuristic. Ablations confirm that removing explicit IVC leads to a performance drop from **0.52 to 0.30**.
- **Novelty of Best-of-N:** We distinguished our contribution from prior value only methods like EMaQ under the tabular setting. Theorem 1 provides the first policy improvement guarantee for the "generate-and-select" mechanism in the general function approximation setting.

# 2. Expanded Empirical Scope (**`Ruko`**, **`ac7s`**, **`dkbj`**)
In response to requests for harder benchmarks to address saturation and broader domains, we added the following evaluations:

- **High-Dimensional Control (D4RL Adroit):** On *Hammer* and *Door* tasks, MFP achieves a **+14 point** improvement over the strong baseline QC.
- **Visual Manipulation:** On the pixel-based *Puzzle-4x4*, MFP achieves a success rate of **0.75** compared to the baseline's 0.47, demonstrating robustness in visual task.
- **Imitation Learning:** Validated on Behavioral Cloning as requested by **`ac7s`**; MFP matches a 20-step baseline's performance using only **1-step inference**.

# 3. Efficiency Verification (**`Ruko`**, **`aBKZ`**)

- **Training Efficiency Verification**: Our MFP achieves highest success rate and fastest training speed. This superior training efficiency stems directly from our one-step action generation, which eliminates the expensive iterative sampling process required by prior multi-step flow policies.
- **Inference Efficiency Verification**: We validated the computational feasibility for real-time control loops by benchmarking inference on a consumer-grade CPU-only environment. It achieves ~10ms latency, comfortably satisfying the strict timing requirements of high-frequency physical control loops.
- **Ablation on Inference Step**: We evaluated the baselines with varying inference steps to demonstrate the trade-off between efficiency and performance. Reducing the number of inference steps for the baselines results in a catastrophic performance drop.

# Conclusion
We have provided specific theoretical justifications to resolve the ambiguity raised by the dissenting reviewer and supplements the paper with the requested high-dimensional and visual experiments to verify robustness and scalability. We respectfully request the AC’s consideration of the reviewer feedback and substantive rebuttal improvements. Thank you for time, effort and professionalism!

---

### Meta-Review · Area_Chair_X1FZ · 2026-01-04

**Summary:**

The reviewers generally agree that the paper presents a novel and theoretically well-motivated application of MeanFlow to policy learning in reinforcement learning, particularly fast one-step inference, as well as its clear writing and empirical gains in success rate and speed on robotic manipulation tasks. Meanwhile, the major concerns are insufficient empirical validation, the computational efficiency, and the theoretical assumptions.

**Reviewer Concerns:**

The rebuttal well addresses empirical evaluation (e.g., additional tasks/analysis) and training/inference efficiency.
However, concerns regarding theoretical assumptions and the incremental nature of IVC’s novelty remain only partially resolved.

**Reviewer Scores:**

dkbj may increase the score from 4 to 6 or keep it. Other three reviewers are likely to keep their original scores (8), as their core concerns were either addressed or the score is high enough for  acceptance.

---

### Decision · Program_Chairs · 2026-01-26

Accept (Oral)